# Lead-free dual-frequency ultrasound implants for wireless, biphasic deep brain stimulation

Qian Wang[1,4], Yusheng Zhang[2,4], Haoyue Xue[1], Yushun Zeng [3], Gengxi Lu [3], Hongsong Fan [2] ✉, Laiming Jiang [1] ✉ & Jiagang Wu [1] ✉

Ultrasound-driven bioelectronics could offer a wireless scheme with sustainable power supply; however, current ultrasound implantable systems present critical challenges in biocompatibility and harvesting performance related to lead/lead-free piezoelectric materials and devices. Here, we report a lead-free dual-frequency ultrasound implants for wireless, biphasic deep brain stimulation, which integrates two developed lead-free sandwich porous 1-3-type piezoelectric composite elements with enhanced harvesting performance in a flexible printed circuit board. The implant is ultrasonically powered through a portable external dual-frequency transducer and generates programmable biphasic stimulus pulses in clinically relevant frequencies. Furthermore, we demonstrate ultrasound-driven implants for long-term biosafety therapy in deep brain stimulation through an epileptic rodent model. With biocompatibility and improved electrical performance, the lead-free materials and devices presented here could provide a promising platform for developing implantable ultrasonic electronics in the future.

Deep brain stimulation (DBS) has emerged as an effective treatment strategy for a wide range of neurological conditions, such as Alzheimer's disease[1,2], Parkinson's disease[3,4], essential tremor[5,6], and epilepsy[7–9]. DBS technology involves surgically implanting electrodes into specific areas of the brain that deliver programmable electrical pulses to modulate abnormal brain discharges, thereby ameliorating or treating disease[10–13]. The power supply for DBS, however, remains a fundamental challenge to be addressed. Conventional external power supplies use transcutaneous or percutaneous wires, which are inconvenient, easily damaged, and susceptible to infection, especially in long-term applications[14–16]. An alternative is to integrate the battery with the implants, however, regular surgery is required for patients to replace the dead batteries with new ones due to the limited life of batteries, which poses additional risk and a healthcare burden[17–19].

To address these challenges, researchers have devoted efforts to develop wireless power sources, of which ultrasound (US) wireless energy transfer technology has received much attention[20–24]. As a safe, non-invasive tool, US has long been widely utilized in medical applications such as the diagnosis, monitoring, and therapy of various diseases and physical conditions[23,24]. As an emerging technology, US wireless energy transfer technology, which utilizes transmitted US waves to carry energy and programmable information, has been integrated into diverse wireless-powered electronics for functional applications[23]. Compared with other wireless energy transfer technologies (e.g., electromagnetic field, EMF), US is an early stage but promising technology that may offer better performance for small (mm-scale) and deep (several cm) implants because US can achieve better spatial resolution and greater penetration through soft

[1]College of Materials Science and Engineering, Sichuan University, Chengdu, China. [2]National Engineering Research Center for Biomaterials, Sichuan University, Chengdu, China. [3]Alfred E. Mann Department of Biomedical Engineering, Viterbi School of Engineering, University of Southern California, Los Angeles, California, USA. [4]These authors contributed equally: Qian Wang, Yusheng Zhang. ✉e-mail: hsfan@scu.edu.cn; laimingjiang@scu.edu.cn; wujiagang0208@163.com

tissues[25–27], giving it the potential benefits in micro-distributed neurostimulators and deep tissue monitoring applications[23].

Unfortunately, current US implantation systems have deficiencies in biocompatibility and harvesting performance related to lead/lead-free piezoelectric materials as well as associated device design, limiting their general applicability. First, piezoelectric materials, due to their excellent electromechanical coupling efficiency, are the key materials for the development of US wireless energy transfer technology[25,28–30]. However, current US transfer systems primarily rely on lead-based piezoelectric materials (e.g., lead zirconate titanate, PZT) as acousto-electric coupling elements[31,32], bringing concerns about the harm to the body, especially being used as the piezoelectric ultrasonic energy harvester (PUEH) to be implanted in the body. Consequently, there remains an urgent challenge for developing lead-free piezoelectric materials and implants that should be biocompatible and high performance to ensure harmlessness to the human body during long-term implantation. Recently, more and more researches on lead-free piezoelectric ceramics are being reported, among which (K,Na)NbO$_3$ (KNN) is one of the most promising alternatives to lead-based counterparts owning to their favorable piezoelectric coefficient ($d_{33}$)[33,34]. The most important parameters to measure the efficiency of PUEH are piezoelectric voltage constant $g_{33}$ $\left( = \frac{d_{33}}{\varepsilon_{33}^T \varepsilon_0} \right)$ and harvesting figure of merit (FOM) $d_{33} \times g_{33}$[35,36]. The key to improving US harvesting performance is to obtain a high piezoelectric constant $d_{33}$ while lowering the dielectric constant $\varepsilon$, yet a high $d_{33}$ always accompanies a high $\varepsilon$ in KNN ferroelectrics. A series of initiatives are therefore required to decouple the $d_{33}$ from $\varepsilon$, namely to reduce $\varepsilon$ without sacrificing the current superior $d_{33}$[37–39]. Secondly, all clinically approved electrical neurostimulation therapies utilize biphasic stimulus pulses with "charge balance" to passively or actively charge and discharge the electrodes during each cycle. This is because, for monophasic stimulation with charge imbalance, the accumulated charges on the stimulating electrodes may lead to electrolysis, which can cause tissue damage or electrode damage[40,41]. However, current US implants typically deliver monophasic stimulus pulses in the absence of complex analog and digital circuits due to single-frequency operation[23,31,42]. Whereas, the use of complex analog and digital circuits poses challenges to device miniaturization due to the deployment of multiple components, rendering it unfavorable for implantable applications. Hence, the development of compact, US-induced deep brain stimulators with biocompatible high-performance lead-free piezoelectric materials and the capability to deliver biphasic stimulus pulses is significant.

To address these requirements, we present a strategy for lead-free piezoelectric composite engineering and device integration to (i) manufacture a biocompatible and highly piezoelectric lead-free sandwich porous 1-3-type (SP-1-3) composite structure with multilayered piezoelectric micro-rods embedded inside an epoxy matrix and (ii) use these lead-free composites to create a flexible biphasic US implant (f-BUI) for DBS, which integrates two PUEHs with different resonant frequencies (1 MHz and 3 MHz) on a flexible printed circuit board (F-PCB) and is then implanted into the rat's brain as a deep brain stimulator. An external dual-frequency US transducer is also prepared for multi-source transmissions. Compared to the originally synthesized KNN piezoelectric materials, the lead-free SP-1-3 composites feature more than a threefold enhancement in $g_{33}$ (19–61.4 × 10$^{-3}$ V m N$^{-1}$) and nearly a twofold increase in harvesting FOM (9120 to 17806 × 10$^{-15}$ m$^2$ N$^{-1}$), which are much higher than that of most other lead-free piezoelectric materials and even some lead-based counterparts[43,44]. The PUEH based on SP-1-3 composite showed a superior energy harvesting performances, reaching the average charging power up to 2286.62 μW. Of importance, due to the dual-frequency design, the f-BUI enables biphasic stimulus pulses through externally programmable multi-frequency US excitation, which was

eventually implanted in the rat's brain for wireless therapeutic DBS to successfully suppress seizures and proved to be biocompatible in the long term. Collectively, the developed lead-free piezoelectric materials and implants could offer not only a promising strategy for biocompatible ultrasonic electronics but also an insight into other implantable device applications.

## Results

### Design and working principle of the f-BUI

Figure 1a shows a conceptual illustration of the f-BUI as an implantable device for DBS. The implant integrates two lead-free SP-1-3 piezo-elements (3 mm × 3 mm footprint) for acoustic-electrical coupling, two pairs of rectifiers and transistors for alternating current-direct current (AC-DC) conversion, and a stimulating electrode in a soft, thin F-PCB (~100 μm board thickness). The US beamlines are programmed and delivered via an external dual-transducer (Supplementary Fig. 1a, b), which is portable and requires no physical connection to the implanted f-BUI. A medical ultrasonic gel is used as a coupling medium between the dual transducer and the f-BUI for efficient acoustic transmission and coupling. Upon being impacted by the US beamlines, the lead-free SP-1-3 piezo-elements will vibrate and generate electrical charges in response to applied acoustic pressure.

The key advances relative to the f-BUI for a wireless deep brain stimulator include (i) the implementation of high-performance lead-free piezo-elements with sandwich, porous, 1-3-type composite structures, which remarkably enhanced biocompatibility and piezo-electricity through multilevel structural engineering (discussed in more detail below). (ii) A dual-channel US system to produce biphasic stimulus waveform through the manipulation of US frequency. Previous piezoelectric stimulators were mostly monopolar stimuli[23,31,42,45]. However, a biphasic stimulus is usually required for biomedical stimulators, especially for some high-frequency stimulations, since its charge balance waveform reduces charge buildup and undesirable electrochemical reactions on the electrode surface[41,46,47]. To achieve this, we employed two sandwich porous 1-3-type composite-based piezoelectric ultrasonic energy harvesters (SP-1-3 PUEHs) with different resonant frequencies (PUEH-1: 1 MHz, PUEH-2: 3 MHz) and connected them inversely to a pair of rectifiers and transistors, respectively, and then programmed the triggered US (frequency and phase) to achieve biphasic stimulation. When the f-BUI is implanted into the mouse skull, the programmed trigger signals first apply to the dual transducer to emit multi-frequency US beamlines, and then the implanted device will accordingly generate "charge-balanced" biphasic pulses for DBS.

The as-assembled f-BUI is shown in Fig. 1b, c, highlighting its flexible, soft mechanical properties and lightweight electronic design comparable to previously reported brain implantable devices[31,48,49], with an overall weight of 0.45 g and an overall dimension of 7.3 mm in width, 13.6 mm in length and 2 mm in thickness, making it suitable for brain implantation in rodent models (Supplementary Table. 1). The implant is finally encapsulated in soft silicone elastomer (Ecoflex-0030, 60 kPa in Young's moduli; Fig. 1d), providing soft and conformal contact between the stimulator and the skin located skull region (Fig. 1e)[50], thus allowing for comfortable implantation as well as efficient transmission and receiving of US waves. Furthermore, acoustic harvesters usually require special angles related to the ultrasonic beam to achieve optimal performance. Therefore, the acoustic field optimization needs to place primary emphasis on the challenges of transmitter-receiver alignment, especially for dual-frequency operation in this work. On the one hand, the external dual-transducer was also designed and fabricated with a soft connection (silicone rubber), providing advantages for maintaining a good angle with the f-BUI (Supplementary Fig. 1c–f). On the other hand, the dual plane-wave acoustic field is employed in our system. Compared to a typical focused acoustic field, this plane configuration features a larger

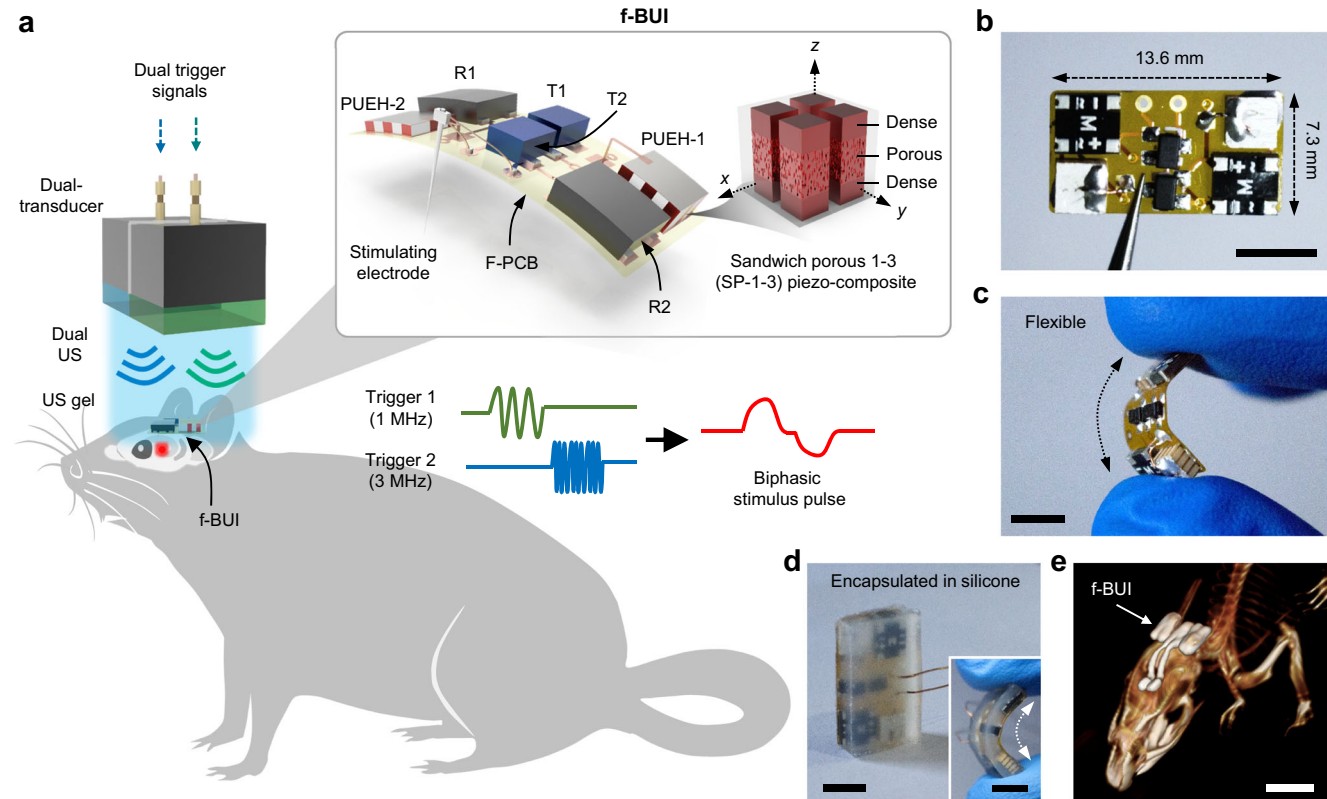

**Fig. 1 | Design and working principle of the f-BUI. a** Schematic diagram of the f-BUI implanted in a rat brain. The f-BUI integrates two SP-1-3 composites with different resonant frequencies on F-PCB, which is induced by a dual transducer to produce biphasic waveform for DBS. The inset shows the schematic layout of the f-BUI. R1 and R2 are rectifiers. T1 and T2 are transistors. PUEH-1 and PUEH-2 are piezoelectric ultrasonic energy harvesters. **b, c** Optical image of the f-BUI before encapsulated in the silicone and demonstration of the device flexibility. Scale bars are 5 mm. **d** Optical image showing the f-BUI encapsulated in silicone. The inset shows the flexibility of the encapsulated device. Scale bars are 5 mm. **e** Micro-CT 3D rendering image of a rat brain after implantation of f-BUI on day 30. Scale bar is 10 mm.

positional tolerance, facilitating the alignment of the two-channel transmitter and harvester in this work (Supplementary Fig. 2).

### Enhanced electrical performance in lead-free SP-1-3 piezo-elements due to multilevel structural engineering

As the core functional component of f-BUI, the piezo-element converts wireless US-induced mechanical vibration into electricity, the coupling efficiency of which can be emphasized by strategic multi-level micro-structure engineering. In terms of the sensitivity of PUEHs, it is determined by the voltage ($V$) produced in the piezo-elements under US-induced mechanical stress ($\Delta\sigma$) and can be quantified by[51]

$$V = \frac{d_{33}}{\varepsilon_{33}^T} \times t \times \Delta\sigma \tag{1}$$

where $t$ is the thickness and $\varepsilon_{33}^T$ is the permittivity. The corresponding produced energy ($E$) across the area ($A$) is calculated from[51]

$$E = \frac{1}{2} \times \frac{d_{33}^2}{\varepsilon_{33}^T} \times A \times t \times \Delta\sigma \tag{2}$$

Thus, it is essential to optimize $g_{33}$ ($= \frac{d_{33}}{\varepsilon_{33}^T\varepsilon_0}$) and harvesting FOM ($= d_{33} \times g_{33}$) to improve the energy harvesting capability of the device, whereas the primary strategy is to achieve a high piezoelectric constant while reducing the dielectric constant.

A multilevel structural engineering that combines sandwich porous (dense/porous/dense) and 1-3-type composite (ceramic/epoxy) structures is exploited for the preparation of lead-free piezoelectric composite elements for f-BUI, as schematically illustrated in Fig. 2a. First, conventional solid-state process combined with burn-out polymer spheres method was employed to prepare the sandwich structured ceramic substrates, where a porous ceramic layer formed between two dense layers and polystyrene (PS) microspheres were used as pore-formers (Supplementary Fig. 3). Phase boundary engineered 0.95(K,Na)(Sb,Nb)O$_3$-0.05(Bi,Na)ZrO$_3$-0.2%Fe$_2$O$_3$ (abbreviated as KNNS95) ceramics with R-O-T phase coexistence (R: 24.8%, O: 40.3%%, and T: 34.9%) exhibiting high piezoelectricity ($d_{33} \approx 480$ pC N$^{-1}$) are synthesized as the piezoelectric matrix (Fig. 2b, Supplementary Fig. 4, and Supplementary Table 2). Although dense ceramics exhibit superior piezoelectric constant, meanwhile, they suffer from high dielectric constants and hence poor energy harvesting performance. Currently, the employment of porous ceramics fabricated by introducing air with low dielectric constant as the second phase is a common strategy to enhance the energy harvesting performance[36,38]. Nevertheless, porous structure leads to a lower electric field strength to which the ceramics are subjected, thus potentially leading to incomplete domain inversion and insufficient and limiting their application in some high fields[37]. Additionally, impurities are more easily infiltrated due to the excessive holes in the ceramic surface, for example, we observed the infiltration of silver electrodes in the porous ceramic cross-section (Supplementary Fig. 5). Consequently, we proposed a sandwich-porous structure and optimized layer thickness ratio as well as the diameter and fraction of PS. X-ray diffraction (XRD) data and temperature-dependent dielectric constant ($\varepsilon_r$-T) profiles of these sandwich ceramics with different pore sizes are shown in Fig. 2c and Supplementary Figs. 6 and 7, indicating that the composite ceramics maintain the R-O-T phase coexistence ensuring high piezoelectricity after the introduction of

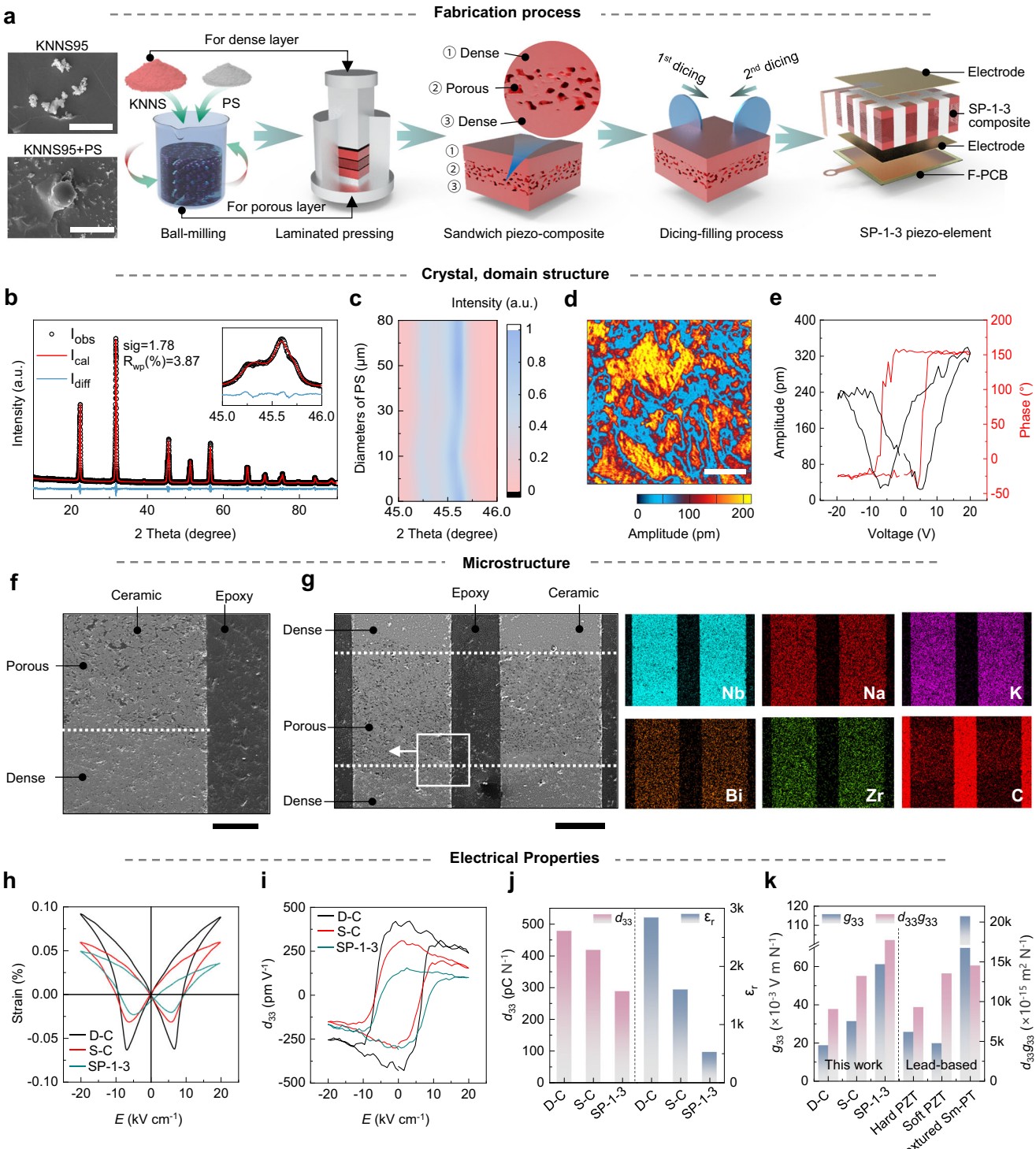

**Fig. 2 | Fabrication and characterization of lead-free SP-1-3 piezo-elements.**
**a** Schematic diagram of the fabrication process of SP-1-3 piezo-elements. The left insets show SEM images of the KNNS95 precursor (top) and a mixture of KNNS95 precursor and polystyrene (bottom).Scale bars are 10 μm (top) and 50 μm (bottom). **b** Rietveld refinement of macroscopic structure of dense KNNS95 ceramics. The inset shows expanded part at the 2Theta range of 45°−46°. **c** Room temperature XRD patterns of piezo-composites with different diameters of PS measured at the 2Theta range of 45°−46°. **d**, **e** Amplitude (**d**) and SS-PFM curves (**e**) of dense KNNS95 ceramics. Scale bar is 1 μm. **f** Local SEM of SP-1-3 composite, showing the

dense, porous, and epoxy resin portion in the composite. Scale bar is 500 μm. **g** SEM image of SP-1-3 composite and corresponding element mapping. Scale bar is 200 μm. Experiments in **f**, **g** were repeated three times with similar results. **h**, **i** Bipolar strain curves (**h**) and $d_{33}$-$E$ loops (**i**) of dense KNNS95 ceramics (D-C), sandwich ceramics (S-C), and SP-1-3 composites. **j** Variation tendency of $d_{33}$ and $\varepsilon_r$ of D-C, S-C, and SP-1-3 composites. **k** Comparison of $g_{33}$ and $d_{33} \times g_{33}$ values between lead-free piezo-materials in this work and lead-based counterparts in other works[55]. Source data are provided as a Source Data file.

the porous and sandwich structures[33,52], but with a significant decrease in $\varepsilon_r$. In addition, PFM measurements were employed to observe the domain structure in the cross-section of dense and sandwich ceramics (Fig. 2d and Supplementary Fig. 8), both of which possess a disordered distribution of irregular nanodomains and whose ease of switching determines ceramics excellent piezo-electricity. The amplitude curves and phase loops of both show typical butterfly-shaped amplitude profiles as well as nearly rectan-gular phase loops (Fig. 2e and Supplementary Fig. 8d), with phase contrasts (Δphase) exceeding 180°, indicating sufficient domain switching and further suggesting that porosity does not lead to a significant decrease in piezoelectricity[53]. Consequently, an optimal $d_{33} \times g_{33}$ value for the sandwich ceramics was $13,272 \times 10^{-15}$ m$^2$ N$^{-1}$ with a layer-thickness ratio of 1:1:1, a PS diameter of 50 μm, and a fraction of 9 wt.% (Supplementary Fig. 9).

To further enhance the energy harvesting performance of the lead-free piezo-elements, a multiple dicing-and-filling technology was utilized to manufacture sandwich porous 1-3 (SP-1-3) piezo-compo-sites, which are composed of periodically configured sandwich KNNS95 piezo-micropillars embedded in a passive polymer matrix (epoxy, EPO-TEK 301, dielectric constant ~4), with the pillar width of 400 μm, a kerf of 200 μm, and a piezoelectric volume fraction of ≈45% (Fig. 2f, g). Electric field- strain (E-S) curves (Fig. 2h), polarization-electric field (P-E) hysteresis loops (Supplementary Fig. 10), and $d_{33}$-electric field loops (Fig. 2i) were studied. The epoxy polymers are without ferroelectric polarization, thus their effect somewhat reduces the total ferroelectricity, piezoelectricity and strain of the SP-1-3 composites[25,54]. Similarly, porousness reduces the ferroelectricity of the ceramics. The ferroelectric and strain properties of the sandwich ceramics with varied proportion and diameters of PS are shown in Supplementary Figs. 11–14. It is noteworthy, however, that $\varepsilon_r$ decreases faster than $d_{33}$ (Fig. 2j and Supplementary Table. 3), with both $g_{33}$ (~$61.4 \times 10^{-3}$ V m N$^{-1}$) and $d_{33} \times g_{33}$ ($17806 \times 10^{-15}$ m$^2$ N$^{-1}$) values increas-ing remarkably in such SP-1-3 composites, even superior to some textured lead-based ceramics (Fig. 2k)[55].

## Wireless performance validation of f-BUI under ultrasonic excitation

The f-BUI enables wireless energy transfer via acoustic approach: converting mechanical energy into electrical energy. Figure 3a and Supplementary Fig. 15 show a schematic illustration of the measure-ment setup in a water tank, where an external dual transducer trans-mits US to the f-BUI and the f-BUI, composed of two SP-1-3 piezo-composites, accepts US and converts it to electricity. The finite ele-ment analysis (FEA) was performed to simulated the piezo-potentials induced under the same US-excitation (Fig. 3b), showing that the average piezo-potential generated in SP-1-3 composite is remarkably higher than those in D-C and S-C materials due to the enhanced har-vesting performance.

To experimentally verify the feasibility and accuracy of US-induced energy transfer, we measured the outputs of f-BUI and first compared the waveforms of the trigger signals and unrectified output signals (Fig. 3c and Supplementary Fig. 16). For example, under the excitation of 1-MHz US pulses with repeat cycles of 200, the induced signal in PUEH-1 shows a equal pulse length (~200 μs) and a consistent repetition period (~1 μs; Fig. 3c). Additionally, a typical time delay (~15 μs) and a ringdown phase are observed in the output waveform due to US traveling and f-BUI vibration attenuation. The same phe-nomenon is observed in PUEH-2 under the excitation of 3-MHz US pulses (Supplementary Fig. 16). These results validate that the output signals are induced by the ultrasonic waves transmitted from the dual transducer. Meanwhile, the pulse amplitude, pulse length, and pulse frequency of the output signals can be flexibly programmable via the external dual transducer, thus enabling an on-demand therapeutic platform for DBS (Supplementary Figs. 17–19).

Since the US transducer only generates alternating current (AC), however, the current needs to pass through the f-BUI rectifier module to be converted to direct current (DC) (Fig. 3d). The change of output voltage and instantaneous power of f-BUI delivered to the loads were estimated by measuring the rectified voltages under varied load resistance ranging from 20 Ω to 50 kΩ (Fig. 3e). The voltage increased rapidly first and then saturated (~17.5 V, >2 kΩ), reaching a maximum instantaneous power of ~0.6 W at 150 Ω, where the output amplitude is 9.5 V. The rectified DC outputs were then stored directly in different energy storage capacitors (470 and 1000 μF) for evaluated the average charging power ($P_{ave}$), which is given by the formula[56]

$$P_{ave} = \frac{CV^2}{2t} \quad (3)$$

where $t$ is the charging time, $V$ is the increased voltage, and $C$ is the capacitance. For example, the voltage stored in 1000 μF capacity increased by 11.1 V in 25 s (Fig. 3f). Consequently, the average charging rate is 444 μC S$^{-1}$ and the $P_{ave}$ reaches up to 2465.4 μw, which are far superior to previously reported lead-free PUEHs and even compares favorably with lead-containing counterparts (Supplementary Tab. 4). In addition, we evaluated the power transfer efficiency of our ultrasonic energy transfer system under 1-MHz and 3-MHz operations, respectively. For the 1-MHz channel, the system exhibited a power transfer efficiency from the acoustic power (617 mW) at the implant's piezo surface to the implant's electrical output power (~156 mW) of ~25.3% and an end-to-end efficiency of ~7.8%, defined as the ratio of the implant's electrical output power to a consumed power (2 W) of the external transmitting transducer. For the 3-MHz channel, under the same consumed power, the system exhibited a power transfer efficiency ~22.5% and an end-to-end efficiency of ~6.5%. The 3-MHz channel presents a lower efficiency than 1-MHz one because the higher the frequency, the faster it attenuates in the propagation media. Although the two channels present different transfer efficiencies, we can achieve a balance between the two channels by adjusting the input power of the external transmitting transducer.

To determine the usability of f-BUI for implants and the effect of tissue thickness on the output signal, we performed ex vivo experi-ments and selected porcine tissue as the medium to mimic the implantation environment. When the f-BUI was placed under porcine tissue, the output voltage decreased with the increase in tissue thick-ness (0, 5, 10, 15, and 20 mm; Fig. 3g and Supplementary Fig. 20), which was a result of ultrasonic power attenuation. Even so, at a thickness of 20 mm, the output voltages under 1 MHz- and 3 MHz-US excitations remained at ~80% and ~70%, respectively. Generally, subcutaneous implantation depths are <10 mm[22], therefore both frequencies of f-BUI work well for implantation applications. Furthermore, the effect of the acceptance angle on the ultrasonic energy transmission is explored (Supplementary Fig. 21). The PUEH has a wide acceptance angle, with piezoelectric voltage harvested varying between 58% and 100% of maximum value over a 30° angular misalignment of the harvester relative to the acoustic beam. Also, we designed and prepared the external plane-wave dual-transducer with a soft connection, providing advantages for maintaining a good angle with the f-BUI (Supplemen-tary Figs. 1 and 2).

Biphasic stimulation is practiced in most bio-stimulators because its charge-balancing waveform reduces charge buildup and undesir-able electrochemical reactions on the electrode surface[40,46]. To pro-duce effective biphasic stimulation within the therapeutic window (100–200 Hz), two lead-free SP-1-3 PUEHs with different resonance frequencies (1 MHz and 3 MHz) were deployed (Supplementary Fig. 22) and connected them inversely to a pair of rectifier modules, respec-tively, and eventually to the same stimulation electrode (Fig. 3h). Transistors prevent currents generated by one SP-1-3 PUEH from propagating through the circuit attached to the other PUEH, ensuring

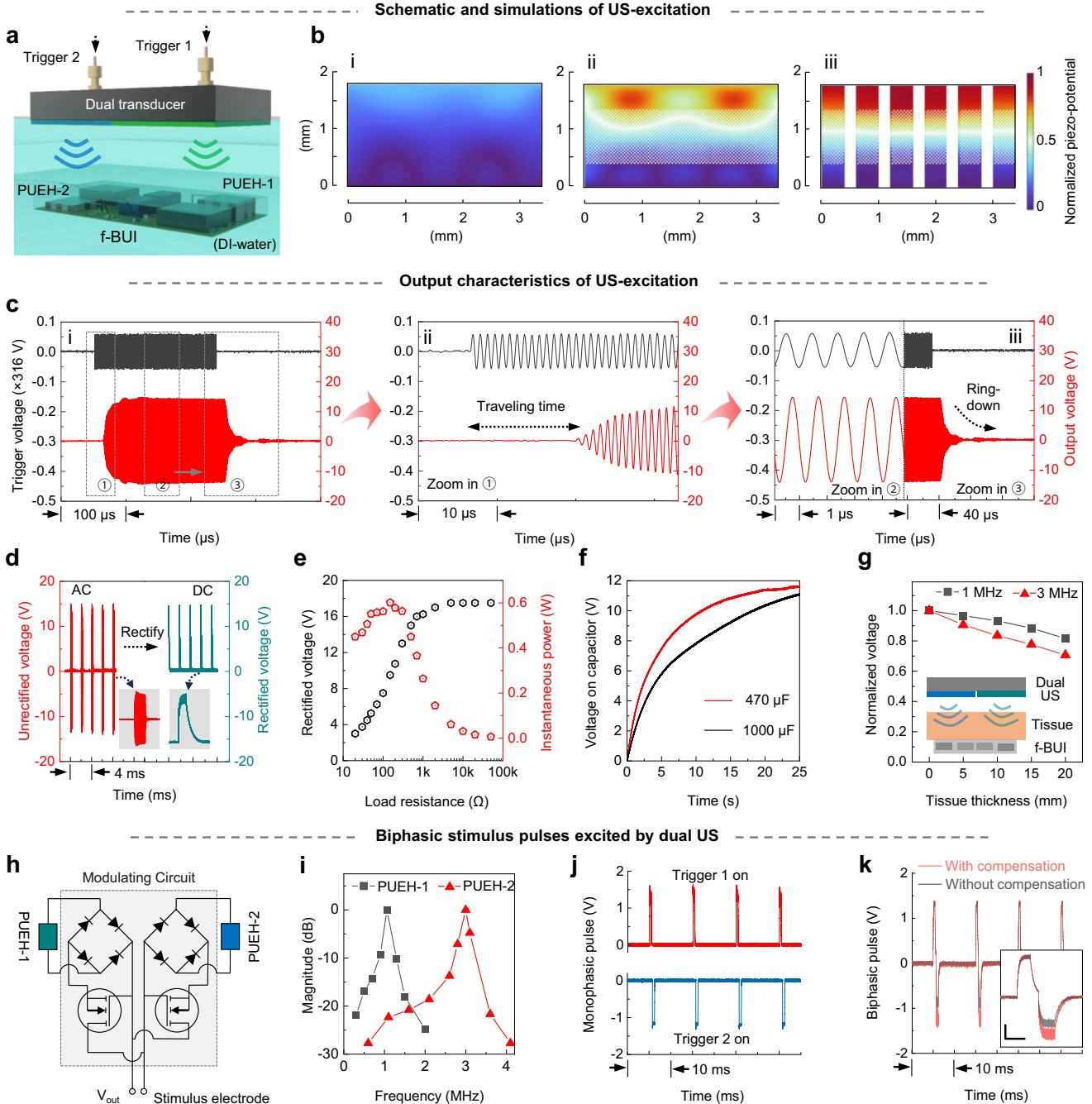

**Fig. 3 | Simulations and measurements of US-induced electrical outputs of f-BUI. a** Schematic diagram of the experimental setup in a water tank. **b** Simulated piezo-potential distributions inside dense ceramics (i), sandwich ceramics (ii), and SP−1-3 composites (iii). **c** Trigger signals to the US transmitter and the output signal from PUEH-1, driven by 1 MHz sine waves. The output signals feature a time delay (ii), frequency consistency (iii, left), and ringdown phase (iii, right). **d** Output voltage amplitudes generated by f-BUI before (left) and after (right) rectification. The insets show the expanded AC (left) and DC (right) signals. **e** Output voltage and the instantaneous power at different load resistances. **f** Stored voltage in capacitors (470 and 1000 μF). **g** Normalized output voltages of the f-BUI under porcine tissues with different thicknesses (0, 5, 10, 15, and 20 mm). The inset shows schematic

diagrams of the ex vivo porcine experimental setup. **h** Schematic of the circuit design in f-BUI to generate biphasic stimulus waveform. **i** Voltage magnitude of two piezo-harvesters with 1 MHz and 3 MHz resonant frequencies when induced by USs of different frequencies. **j** Two inverted monophasic stimulus pulses induced by two single trigger signals. Top, trigger 1 on. Bottom, trigger 2 on. **k** Biphasic stimulus pulse generated by the f-BUI when both trigger signals are turned on simultaneously. The charge imbalance due to different channel transmission efficiencies can be adjusted by compensating for the 3-MHz trigger signal. The inset shows the enlargement of biphasic waveforms. Scale bars are 0.5 ms and 0.5 V. Source data are provided as a Source Data file.

that only one half of the circuit is in operation at a time. In addition, due to the two resonance frequencies setting in the f-BUI, the result shows that with one frequency as the US frequency, the reception strength of the off-resonance attenuates severely (Fig. 3i and Supplementary Fig. 23). The results show that dual-frequency US can

individually excite elements without interacting with each other. When a single trigger signal (trigger 1 or trigger 2) is turned on, we can only observe monopolar positive or negative pulses generated by f-BUI (Fig. 3j). As long as the trigger 1 (frequency: 1 MHz, pulse length: 400 μs, pulse period: 100 Hz) and trigger 2 (frequency: 3 MHz, pulse length:

400 μs, pulse period: 100 Hz, pulse delay: 200 μs) are turned on simultaneously at their resonance frequencies (Supplementary Fig. 24a), the biphasic pulse can be noticeably observed (Fig. 3k), whereas charge imbalance pulses are also observed due to the difference in channel transmission efficiency (Fig. 3k, gray line). For both channels, although their corresponding transmitters and receivers are of the same material and construction design, the difference in ultrasonic frequency does result in varying transmission efficiency, for example, one of the reasons for this is the greater attenuation of 3-MHz ultrasound (Fig. 3g). To effectively compensate for the imbalance in charge, we increased the trigger signal for the 3-MHz channel. From the test results, we can see that by compensating the trigger 2 signal at the input (boosting ~22%) (Supplementary Fig. 24b), the output pulse can be relatively balanced (Fig. 3k, red line), thus providing better assurance for subsequent stimulation applications. We could also modulate the amplitudes of biphasic stimulation waveform by adjusting the trigger voltages of both channels simultaneously (Supplementary Fig. 25). In addition, the pulse amplitude attenuates significantly once they are switched to the off-resonance operation (Supplementary Fig. 26).

## DBS therapeutic benefits of f-BUI for seizure suppression using animal models in vivo

To validate our f-BUI system, we conducted a series of in vivo studies to examine the efficacy and robustness of our developed device in delivering therapeutic DBS for seizures. DBS is a priority neuromodulatory treatment, particularly biphasic electrical stimulation, which effectively maintains the charge injection balance and prevents tissue damage. Thus, the f-BUI system based on lead-free, dual-frequency US implants is a promising candidate for DBS in epilepsy treatment. As shown in Fig. 4a, the dual transducer can generate dual US beams penetrating the tissue, which is capable to be received by f-BUI to trigger wireless electrical stimulation. First, the device was subdermally implanted in a rat brain through the skin, according to the standard procedure (Fig. 4b and Supplementary Fig. 27; surgical procedure is detailed in "Methods"). Furthermore, X-ray images were performed to examine the spatial location of the implanted device and electrode, as well as the surrounding tissue conditions, as shown in Fig. 4c and Supplementary Fig. 28. The results indicated that the components of the device remained in their original positions over the course of 30 days following surgery, with no signs of degradation in functionality or appearance, providing stable environmental conditions for epilepsy treatment (Supplementary Movie 1). Furthermore, we performed experiments using a phantom skull to study the feasibility of device implantation in the human brain, which demonstrated the f-BUI is smaller enough than the current DBS device and is an attractive option for potential applications to the human brain for the treatment of neurological or psychiatric disorders (Supplementary Fig. 29).

To further investigate the DBS antiepileptic effects of the f-BUI system, a 4-aminopyridine (4-AP)-induced seizure model was established, which can generate table seizure-like events occurred conditions regarding evoked epileptiform burst activity morphology and frequency over several hours[57–59]. In our experiments, electrocorticography (ECoG) signals recorded while US+f-BUI stimulation in the Sham group (Supplementary Fig. 30a) showed that the amplitude of biphasic electrical stimulation artifacts is much larger than the local field potentials as shown in previous reports[60,61], which made an interference on the LFP recording. Meanwhile, the ECoG signals recorded before and after US+f-BUI stimulation showed that the normal brain activity was similar with the signal before electrical stimulation (Supplementary Fig. 30b), demonstrating that US+f-BUI stimulation have no adverse effect on the normal brain activity, which was beneficial from the biosecurity afforded by biphasic electrical stimulation. Therefore, we recorded the ECoG signals before and after

stimulation, and there were four groups: the sham group, the seizure group, the US alone (US) group, and the US-induced f-BUI (US+f-BUI) group. As shown in Fig. 4d, when the seizures reached stage 5 according to the Racine score, representative ECoG signals record and related time-frequency spectrum in the seizure, US, and f-BUI groups showed typical sharp epileptic waves with greater amplitudes compared to the sham group. After 15 min of US stimulation (0.2 MPa, 100 Hz pulses), the amplitude of the epileptic waves in the US group decreased, similar to previous reports[62,63], and the result suggests that slight epilepsy inhibition may be achieved by the plane-wave US generated from the dual transducer. However, as a comparison, owing to electrical stimulation generated by f-BUI under US-induced stimulation, sharp epileptic waves significantly decreased and disappeared in the US+f-BUI group, demonstrating that the antiepileptic effect of this system is more effective than that of other groups. Additionally, the seizure severity in the US+f-BUI group was significantly lessened following stimulation as compared to before stimulation, according to the results of the ECoG power spectral density (PSD) (Fig. 4e and f). The integrated area under the PSD curve (Fig. 4g), representing the overall power or strength of the ECoG signals, also confirmed that treatment with US+f-BUI effectively reduced seizure severity. Subsequently, as a result of behavioral observations based on the Racine score, the seizure stage in the US+f-BUI group declined dramatically (Fig. 4h).

We further observed that the rate of death in the 4-AP rat epilepsy model was high if seizures were not inhibited in time. Of note, the survival rate of the US+f-BUI group was significantly higher than the seizure group, demonstrating that the f-BUI system could effectively decrease the mortality of epileptic rats by inhibiting aberrant ECoG signals propagation (Fig. 4i). Meanwhile, 4-AP induced intense and frequent epileptic discharges and damage in both the hippocampus and the cerebral cortex, resulting in the shrinking and loss of hippocampal cells in the CA1 and CA3, as well as hippocampal sclerosis[64–67]. As shown in Fig. 4j and Supplementary Fig. 31, the CA3 region Nissl staining and the number of cells were quantitatively analyzed. It showed no evident difference between the Sham group and Sham+US+f-BUI group, indicating that there was no negative impact of US+f-BUI on the normal state. Furthermore, the architecture and cell number of the CA3 hippocampal regions in the US+f-BUI group was resembling to that in the sham group. Contrary to that, the shrinkage of hippocampal cells in the seizure and US groups increased due to the failure to suppress epilepsy in time. Consequently, although there was still a degree of oscillatory rhythm after US+f-BUI stimulation, the abnormal ECoG signals is significantly lower than the seizure group. The inhibitory effect of this device may be related to other stimulus parameters, and there are still many parameters, which should be to optimize to achieve more efficient effect of treatment, such as US intensity and frequency. However, the results have confirmed that the current biphasic electrical stimulation induced by US+f-BUI can effectively reduce seizure severity, showing promising DBS applications in the treatment of epilepsy and other neurological disorders through further research in the future.

## Biological safety verification of lead-free f-BUI

Considering the long-term therapeutic procedure for DBS, the biological safety of lead-free f-BUI must be investigated. The KNN-based SP-1-3 piezoelectric composite, a crucial component of the device, is a promising material for applications in US wireless electronics. As shown in Fig. 5a, leaching liquors of SP-1-3 piezoelectric elements were prepared to investigate cell toxicity. The live/dead cell staining demonstrated that, as time progressed, the proliferation of PC12 cells cultured with the leaching liquors of the SP-1-3 element was similar to that of normally grown cells (Fig. 5b) Meanwhile, the cell proliferation investigated by the CCK-8 assay confirmed that the cytotoxicity of the SP-1-3 element was negligible (Fig. 5c), indicating its superior cytocompatibility.

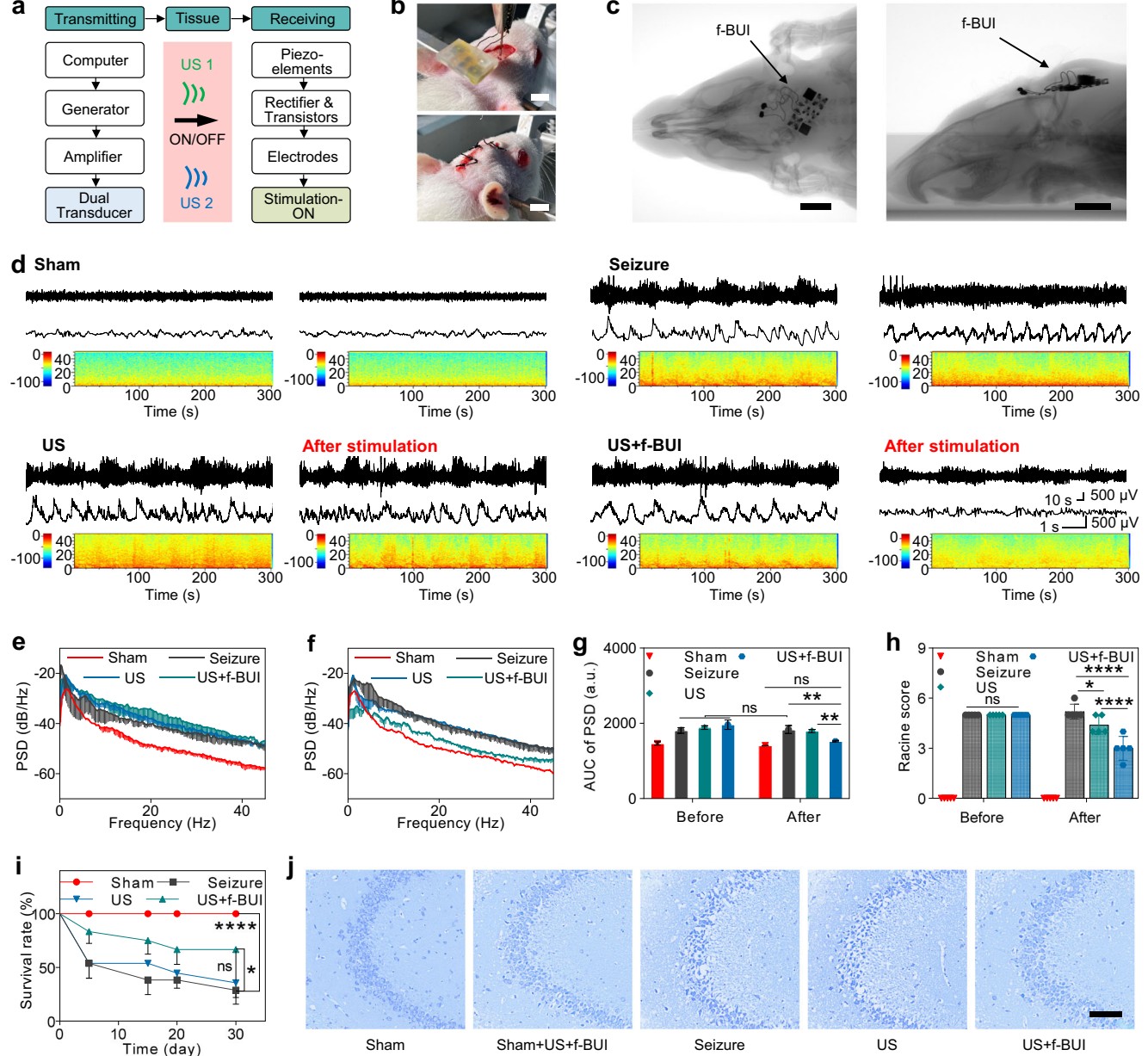

**Fig. 4 | Demonstration of DBS for seizure suppression using f-BUI in vivo.**
**a** Block diagram illustrating the US-induced f-BUI system. **b** Optical images of the implanted procedure of f-BUI. Scale bars are 4 mm. **c** X-ray images of a rat brain after implantation of f-BUI on day 30. Scale bars are 10 mm. **d** Corresponding ECoG signal and spectrum of time-frequency of 4-AP induced epileptic rats in the Sham, Seizure, US, and US+f-BUI groups. **e**, **f** Averaged power spectrums of the different groups before US-induced stimulation (*n* = 3 biologically independent samples) (**e**) and after US-induced stimulation 15 min (*n* = 3 biologically independent samples) (**f**). **g** Aera under curve of the power spectrum before and after US stimulation (*n* = 3 biologically independent samples). **h** The change of Racine score under different

treatment (*n* = 5 biologically independent samples). **i** Survival rate of seizure rats induced by the 4-AP after different treatments on day 30 (*n* = 12 biologically independent samples). **j** Nissl staining in CA3 regions of hippocampus on day 30 (*n* = 3 biologically independent samples). Scale bar is 100 μm. Statistical significance was determined by Two-analysis of variance (ANOVA) with Tukey's multiple comparisons (**g**, **h**) and the Kaplan-Meier method with log-rank test (**i**) (****$p < 0.0001$, ***$p < 0.001$, **$p < 0.01$, *$p < 0.05$, and ns, not significant); data were presented as mean values ± standard deviations (SD); error bars = SD. Source data are provided as a Source Data file.

Furthermore, biological safety of the device in vivo was investigated as shown in Fig. 5d. After 30 days of device implantation, open field test was conducted to examine the spontaneous activities and inquiry behavior of experimental animals in unfamiliar environments. The results showed no significant change in the motion trace or total distance between the Sham and f-BUI groups (Fig. 5e, f). However, after 15 min of US-induced stimulation, the motion distance showed a significant difference in the US+f-BUI group, whereas it was not observed in the US-only group, which demonstrated that device implantation did not detrimentally affect movement and could realize

neuromodulation under US-induced stimulation. Moreover, hematoxylin and eosin (H&E) staining of the skin and major organs (heart, liver, spleen, lung, and kidneys) showed no inflammatory infiltration of the adjacent tissues and no geometric tissue deformations in the f-BUI group compared to the Sham group (Fig. 5g, h). Immunofluorescence staining and quantitative analysis of the implanted site demonstrated that there was no clear inflammatory reaction or surrounding tissue damage after f-BUI implantation (Fig. 5i, j).

Many applications of functional electrical stimulation use charge-balanced biphasic pulses to avoid tissue damage or electrode damage,

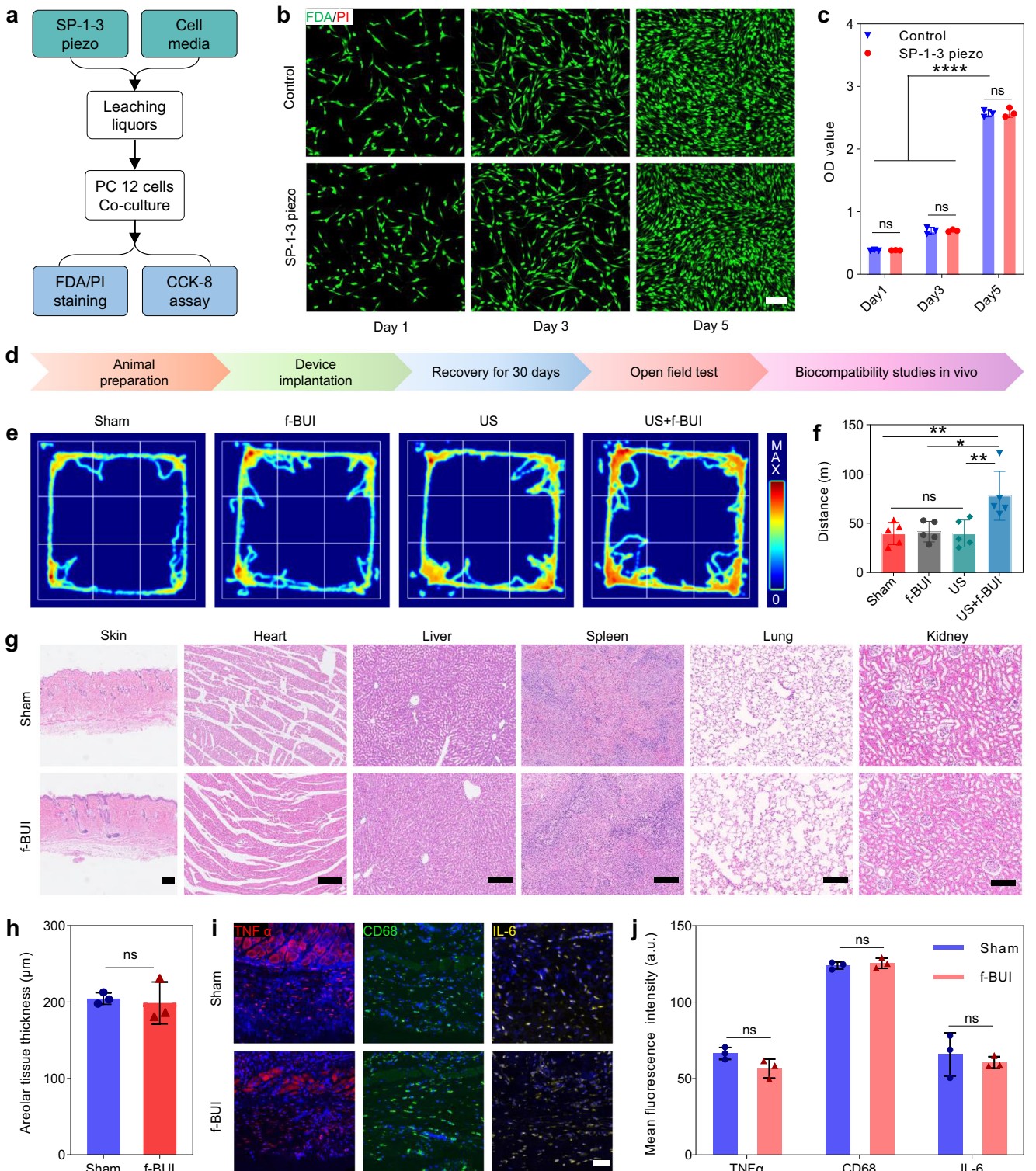

**Fig. 5 | Demonstration of the biological safety of f-BUI. a** Schematic diagram of examining cytocompatibility of SP-1-3 piezoelectric elements. **b** Live (green)/Dead (red) cells staining treated with/without SP-1-3 piezoelectric element leaching liquors on day 1, 3, and 5 (*n* = 3 biologically independent samples). Scale bar, 100 μm. **c** Coresponding cell viability assessed by CCK8 assay (*n* = 3 biologically independent samples). **d** Experimental procedure of f-BUI biological safety in vivo. **e** Representative motion route heatmap in Sham, f-BUI, US, and US+f-BUI group. **f** Total motion distance of SD rats with the different treatment derived from open field test (*n* = 5 biologically independent samples). **g** H&E staining images of skin heart, liver, spleen, lung, and kidney of the rats with/without the implantation of f-BUI (*n* = 3 biologically independent samples). Scale bars are 200 μm.

**h** Quantitative analysis of areolar tissue thicknesses, indicating there are no significant differences between two groups (*n* = 3 biologically independent samples). **i** Representative immunofluorescence images of inflammatory response (TNFα, CD68, and IL-6) of implantation site (*n* = 3 biologically independent samples). Scale bar is 50 μm. **j** Quantitative mean fluorescence intensity of TNFα, CD68, and IL-6 in the implantation site (*n* = 3 biologically independent samples). Experiments in **b**, **g**, **i** were repeated three times with similar results. Statistical significance was determined by One-ANOVA (**f**, **h**, **j**) and Two-ANOVA (**c**) with Tukey's multiple comparisons (****$p$ < 0.0001, ***$p$ < 0.001, **$p$ < 0.01, *$p$ < 0.05, and ns, not significant); data were presented as mean values ± standard deviations (SD); error bars = SD. Source data are provided as a Source Data file.

thus maintaining biocompatibility[41,47]. To verify the benefits of biphasic stimulation, we also conducted electrolysis experiments ex vivo (Supplementary Fig. 32) and stimulation experiments in vivo using a single US channel (output monophasic pulses: 1 ms, 100 Hz, 1.5 V) and a dual US channel (output biphasic pulses: 1 ms, 100 Hz, ±1.5 V), respectively. In electrolysis experiments, where the stimulating electrode was immersed in phosphate buffer saline (PBS), bubbles were observed to be generated on the electrodes during single US operation (Supplementary Fig. 32b), indicating that water is being electrolyzed and it is harmful. However, no significant electrolysis was found during dual US operation (Supplementary Fig. 32c). For stimulation experiments in vivo, the duration of monophasic and biphasic stimulation was set as 30 min per day. After 5 days, we performed histological analyses of the stimulated tissues. Compared to biphasic stimulation, monophasic stimulation increased the extent of the lesion and area of damage, in which monophasic pulses induced a cumulative charge density that created an electrolytic lesion and, consequently, could induce more serious brain tissue damage (Supplementary Fig. 33a, c). Meanwhile, immunofluorescence staining and quantitative analysis of TNF-α, CD68, and IL-6 were performed to further investigate the inflammatory response caused by brain damage, demonstrating that the expression of inflammatory markers in the biphasic stimulation group was significantly lower than that in the monophasic stimulation group (Supplementary Fig. 33b, d). These results indicate that biphasic stimulation can substantially minimize the inflammatory response and tissue damage while still providing effective electrical stimulation. Taken together, all these results collectively indicate the good biocompatibility and biological safety of the lead-free f-BUI system for long-term DBS treatment, which lays the foundation for advancing the practical application of lead-free devices.

## Discussion

In this study, we have developed a flexible, lead-free, dual-frequency ultrasound implants for wireless, biphasic DBS, which mainly integrates two specially designed SP-1-3 lead-free piezo-components on a F-PCB and is ultrasonically powered through a handheld external dual-frequency transducer to generate stable, programmable biphasic stimulus pulses. Such an f-BUI system does not require complex analog and digital circuits or even application-specific integrated circuit (ASIC) and takes all the advantages of strategic lead-free materials engineering and optimal device integration by leveraging the ultrasonic wireless control and power delivery technology. The lead-free SP-1-3 composites that combine porous, sandwich, and 1-3 type composite structures in one unit simultaneously enhance the piezoelectric voltage constant and harvesting figure of merit to $61.4 \times 10^{-3}$ V m N$^{-1}$ and $17,806 \times 10^{-15}$ m$^2$ N$^{-1}$, enabling the device with biocompatibility and superior therapeutic sensitivity. Due to the dual-frequency design, the f-BUI enables biphasic stimulation through externally programmable multi-frequency ultrasonic excitation. Meanwhile, this soft, flexible, and wireless integration enables the mechanical softness of the f-BUI design by eliminating rigid and bulky components such as batteries and large electronic units from the implantable system. Animal experiments demonstrated that the rodent models with epilepticus was successfully therapeutized by using f-BUI for DBS in vivo.

The lead-free piezoelectric materials and dual-frequency US implants presented here could offer insights into the development of future implantable ultrasonic electronics. In addition to the biphasic stimulation functionality demonstrated in this study, such a device layout can also be scaled to more ultrasonic frequency channels (three or more) having a diversity of harvesters for improved reliability and communication with higher data throughput, which could be realized by integrating ASICs with multiple harvesters. Since each channel can be independently manipulated, it is possible to implement arrayed layouts for zoned stimulation of brain tissue, without interfering with each other. Furthermore, it is possible to realize multiple functionalities in a single integrated device, such as one ultrasound channel for information transmission and one for power transmission, thus realizing real-time monitoring and treatment of the human body. Further research would also include control software and hardware optimization, especially to achieve integration with artificial intelligence for programmable adaptive therapy. This requires the implants capable of integrating multimodal sensors and stimulators to enable real-time monitoring and active therapy with minimal physician intervention[68,69]. Although the current design of the f-BUI utilizes a flexible architecture, in future work, materials and device engineering will be further optimized to achieve stretchability or miniaturization of the implants with minimal mechanical burden on the implanted tissue or organ[70]. The application of bioresorbable materials prepared as transient bioelectronics could also be considered to eliminate potential risks during device removal[71]. Since current ultrasound transmitters still rely on lead-based hard PZT materials (e.g., PZT-4 and PZT-8), the future development of alternative lead-free piezoelectric materials with high-quality factor ($Q_m$) for use in transmitters outside the body is also necessary for greater safety. In addition, there are substantial challenges such as skin-transducer coupling that are unresolved. The development of flexible, stretchable, and even bioadhesive external transducers is required to improve skin-transducer coupling and to facilitate long-term wear[72], which involves elaborate design of the electrodes, the matching layer and the coupling agent of the device. Finally, further experiments in large animals and humans are necessary in the future for eventual clinical translation.

## Methods

### Preparation of KNNS95 ceramics (D-C and S-C)

KNN-based ceramic powders with optimized composition were prepared to obtain the D-C and S-C. The lead-free ceramic 0.95(K,Na)(Sb,Nb)O$_3$-0.05(Bi,Na)ZrO$_3$-0.2%Fe$_2$O$_3$ (KNNS95) powders were synthesized first using a conventional solid-state sintering process, with raw materials including K$_2$CO$_3$ (99%), Na$_2$CO$_3$ (99.8%), Nb$_2$O$_5$ (99.5%), ZrO$_2$ (99%), Sb$_2$O$_3$ (99.99%), Fe$_2$O$_3$ (99%) and Bi$_2$O$_3$ (99.999%), purchased from Sinopharm Chemical Reagent Co., Ltd. The detailed preparation process is described in Supplementary Note 1.

### Preparation of SP-1-3 piezo-composites

We used the sintered sandwich-structure ceramics (S-C) to prepare SP-1-3 piezo-composites. The dicing and filling method was first used to manufacture the 1-3-type piezo-composites (SP-1-3) comprising regular-shaped KNNS95 micro-pillar arrays with an epoxy filler (EPO-TEK 301, Epoxy Technology). The composite samples were then mechanically thinned and polished to different thicknesses. Due to the feature of intolerant to high temperatures of epoxy, we give the SP-1-3 composites electrodes by pasting conductive silver paste (E-Solder 3022, Von Roll Isola) on their both sides. Finally, we poled the SP-1-3 composites in silicone oil in a 2.5 kV mm$^{-1}$ DC field at room temperature for 30 min.

### Fabrication of f-BUI and dual transducer

The f-BUI is assembled on a custom F-PCB. Two SP-1-3 piezo-composites (3 mm × 3 mm footprint) with different resonant frequencies (PUEH-1: 1 MHz, PUEH-2: 3 MHz), two rectifiers, and two transistors (R1, R2 and T1, T2) were fixed to the F-PCB by conductive silver paste (E-Solder 3022, Von Roll Isola) sequentially. In the circuit design, the two SP-1-3 piezo-composites are inversely connected to two pair of rectifiers and transistors, respectively. Finally, we made an encapsulation for f-BUI with silicon elastomer membrane (Ecoflex-0030, Smooth-on Inc). The dual-transducer was prepared using two PZT-4 piezoelectric ceramic plates assembled in a 3D-printed frames (see Supplementary Note 2 for detailed preparation process).

## Characterization of piezoelectric materials

The crystal structure of the samples was evaluated by the XRD with Cu Kα radiation (Bruker D8 Advanced XRD, Bruker AXS Inc., USA). Rietveld refinement is performed on Findit and MAUD software. Piezoelectric constant $d_{33}$ was measured by the quasi-static piezo-$d_{33}$ meter (ZJ-3A, Institute of Acoustics, Chinese Academy of Science). The room-temperature dielectric constant was tested using an LCR analyzer (HP 4980, Agilent, Santa Clara, CA). The temperature-dependence of the dielectric constant at different frequencies was tested by using a dielectric spectrometer (Tonghui 2816 A) with a heating rate of 5°C min$^{-1}$. The morphology of both surface and cross section was studied by using a scanning electron microscopy (agellan400, FEI Company), where the samples were first polished and then thermally etched. The room temperature and temperature-dependent ferroelectric and strain properties were tested at 1 Hz by using a ferroelectric analyzer (TF 2000, aixACCT Systems GmbH, Germany). The $d_{33}$-E loops were also detected by the ferroelectric analyzer at a superimposed AC voltage of 25 V in 250 Hz. Domain structures of 2 μm × 2 μm areas were measured using a piezoelectric force microscopy (PFM, MFP-3D, Asylum Research, Goleta, USA) with a driving voltage of 2 V. The local amplitude and phase curves were tested with the SSPFM mode in a voltage of 20 V.

## Measurements of US-induced f-BUI

The input sinusoidal signals with different frequencies were produced by an arbitrary function generator (AFG1062, Tektronix) and then amplified (≈55 dB) by using an amplifier (SSPA1M100M-100N, Sunfire Tech) to trigger the dual transducer, which was constructed of two commercial PZT plates with resonant frequencies of 1 MHz and 3 MHz separately and fixed in a 5D-axis stage. The f-BUI was placed under the dual transducer to receive the ultrasonic signals. Subsequently, the generated output voltages of f-BUI were examined by a digital oscilloscope (TBS 2000 Series, Tektronix). Both the dual transducer and the f-BUI were immersed in a tank with DI water during the measurement. The acoustic pressure emitted by the external ultrasonic transmitters was measured at the surface of the implant piezo in the water tank using a hydrophone probe (NH0200, Precision Acoustics Ltd.).

## Finite element analysis

FEA simulations were conducted using the COMSOL software (COMSOL Multiphysics 5.3a) to model the generation of piezoelectric potential in three kinds of piezoelectric materials (D-C, S-C and SP-1-3 composites) under the same force, as shown in Fig. 3b. Specifically, to study the acoustic-electric coupling, the physical fields including electrostatics, solid mechanics, pressure acoustics as well as the coupled interfaces of the piezoelectric effect were meticulously considered. The physical parameters of water, epoxy, and KNN materials were derived from supplier datasheets and experimental results. In addition, the geometries of the piezoelectric components strictly follows the design requirements.

## Cell culture and toxicity

Rat pheochromocytoma 12 (PC12) cells cultured in high glucose-DMEM medium (Hyclone, China) supplemented with 10% fetal bovine serum (TBD science, China) and 1% penicillin-streptomycin (Hyclone), and were placed in an incubator with 5% $CO_2$ and 95% humidity at 37 °C. For evaluating the cytotoxicity, SP-1-3 element leaching liquors were prepared as ISO 10993-12:2012 (biological evaluation of medical devices) after steam sterilization, which were collected by immersing the material in a DMEM medium with a weight-to-volume ratio of 0.2 g ml$^{-1}$ and incubating them in a cell incubator at 37 °C with 5% $CO_2$ for 72 h before cell counting kit-8 (CCK8) experiment and live/dead cell staining (FDA and PI) on day 1, 3, and 5.

## Animals preparation

Male Sprague Dawley (SD) rats, aged 6–8 weeks, were purchased from Chengdu Dossy Experimental Animal Company (Chengdu, China) and provided sterile food and water and kept on a 12-h light-dark cycle (lights on from 8 A.M. to 8 P.M.). All animal experiments were completed in agreement with the Animal Care and Use Committee of Sichuan University (approve number: SYXK (Sichuan): 2019-189).

## Surgery procedure

For transplantation of the device in vivo, SD rats were anesthetized with sodium pentobarbital (30 mg kg$^{-1}$) and were placed into a stereotaxic frame for surgery. The skull was exposed, and the burr holes were drilled for the implantation of the stimulating electrode at the following coordinates: AP: −1.44 mm, ML: ±1 mm, DV: 6 mm. Then, the device was placed under the head skin and above the skull, and near rat's neck. Micro-CT images were obtained and analyzed using Quantum Image Viewer software (Quantum GX Simple Viewer, PerkinElmer, USA). For the ECoG recording, the recording electrodes were implanted after the transplantation of the device and placed at the following coordinates: AP: −3.5 mm, ML: ±1.2 mm. The electrodes were then protected with dental cement. Rats were given 10 days to recover after surgery before performing further experiments.

## Seizure induction and treatment

4-aminopyridine (4-AP, 5 mg kg$^{-1}$) was used to induce acute seizure model by intraperitoneal injection. Seizure grades were evaluated by Racine scores: Stage 1, immobilization and facial movement; Stage 2, head nodding; stage; Stage 3, unilateral forelimb clonus; Stage 4, rearing with forelimb clonus; Stage 5, falling and jumping; Stage 6, tonic convulsions and hindlimb extension. A Blackrock Neuroport system (Blackrock, USA) was used to measure electrocorticography (ECoG) signals (sampling rate: 30 kHz; bandpass: 1-250 Hz), which were then analyzed using NeuroExplorer software. US-induced stimulation was performed by an external dual-frequency transmitter (1 and 3 MHz), function generators (AFG3252C, Tektronix), and power amplifiers (SSPA1M100M-100N, Sunfire Tech), which generated pulsed acoustic pressure with 100 Hz pulse repetition frequency. Specifically, the dual transducer was aimed at the f-BUI device (based on the micro-CT imaging) after removing the rat's hair and spreading medical ultrasonic gel on the skin and was fixed tightly with a bandage, which formed a head-mounted structure. Medical ultrasonic gel was utilized as a coupling medium for efficient delivery of ultrasound waves between the dual transducer and skin.

## Histological staining

At the 30-day endpoint, SD rats were deeply anesthetized with an intraperitoneal dose of sodium pentobarbital (100 mg kg$^{-1}$) and transcardially perfused with PBS and 4% paraformaldehyde (PFA) in PBS. The major organs, including heart, liver, spleen, lung, kidney, and brain, were sectioned, and keep in PFA for 24 h. H&E and Nissl stainings were performed on paraffin-embedded sections in accordance with the standard protocols. For immunofluorescence staining, the sections were incubated for 12 h at 4 °C in rabbit anti-TNFα (1:500, Abcam), rabbit anti-IL-6 (1:200, Huabio), and mouse anti-CD68 (1:100, Abcam). Then, sections were incubated with the Alexa 555 goat anti-mouse IgG or Alexa 555 goat anti-rabbit IgG. Subsequently, the nuclei were stained with DAPI for 15 min. Finally, the sections were visualized and analyzed using a digital slide scanner (Pannoramic MIDI, 3DHistech Ltd).

## Open field test

The open field test was performed in a black square box (length × width × height: 100 cm × 100 cm × 100 cm) to examine autonomous motion and exploration of rats 30 days after the device was implanted. Individual rats were given 15 min to explore. The motion trace was

automatically scored using the OFT-100 opening experiment system (TechMan Software, Sichuan). Additionally, after one rat completed the test, 75% alcohol solution was used to wipe the box.

## Statistical analysis

Results were shown as the mean value ± standard error of measurements (SEM), unless otherwise noted. GraphPad Prism 9 was used for the statistical analyses, which were followed by one-way ANOVA or two-way ANOVA with Tukey's honestly significant difference test (HSD), Survival rate curve was analyzed by the Kaplan-Meier method with log-rank test. * $p < 0.05$, ** $p < 0.01$, *** $p < 0.001$, and **** $p < 0.0001$, ns $p > 0.05$.

## Reporting summary

Further information on research design is available in the Nature Portfolio Reporting Summary linked to this article.

## Data availability

The data that support the findings of this study have been included in the main text and Supplementary Information. All other relevant data supporting the findings of this study are available from the corresponding authors upon request. Source data are provided with this paper.

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

## Acknowledgements

The work was supported by National Science Foundation of China (U23A20567 and 52202144).

## Author contributions

L.J. and J.W. conceived and designed experiments. Q.W. and L.J. prepared the piezoelectric materials and devices and conducted tests, data collection, and analysis. H.X. helped with device testing. Y.Z. and H.F. conducted in vivo and ex vivo experiments. G.L. and Y.Z. performed simulations by COMSOL Multiphysics. L.J., Q.W., and Y.Z. wrote the manuscript. J.W., L.J., and H.F. supervised the work. All authors discussed and commented on the manuscript.

## Competing interests

The authors declare no competing interests.
