## [Peer Review File · Nature Communications]

REVIEWER COMMENTS

Reviewer #2 (Remarks to the Author):

This paper introduces a wireless and biphasic deep brain stimulator for epilepsy suppression. While the use of ultrasound for energy transfer in electronic stimulation has been reported in several papers, the novelty in this paper lies in the utilization of two ultrasound frequencies to generate biphasic stimulus. However, the advantages of biphasic stimulation need more comprehensive demonstration, and additional experiments should be conducted to compare the results with single-frequency stimulation.

Major comments:

1. The rationale for using two ultrasound frequencies and the advantages of biphasic stimulation in epilepsy inhibition need further clarification. It is advisable to conduct experiments to verify the benefits of biphasic stimulation.
2. The paper states that ultrasound energy is converted to electronic stimulation, but it does not sufficiently address the possibility of ultrasound alone inducing neural activity and epilepsy inhibition. Additional experiments or discussions are needed to distinguish the effects of electronic stimulation from ultrasound.
3. The acoustic field of the ultrasound is crucial for energy exchange efficiency, and this aspect should not be overlooked. The paper should include the characterization and optimization of the acoustic field to improve energy transfer efficiency.
4. The epilepsy model used in the paper is an acute model, and it is uncertain whether this method is applicable to a chronic model, which is more widely accepted in the epilepsy research community.

Minor comments:

1. The abstract requires improvement. Important information is missing, and the novelty of the paper should be more clearly demonstrated.
2. The introduction should emphasize the advances in epileptic inhibition using deep brain stimulation (DBS) and should reference key literature in the field.
3. Please provide the full name and description of SP-1-3 PUEHs when they are first mentioned in the main text.
4. The paper should address the size of the implantable stimulator (7.3 mm width, 13.6 mm length, 2 mm thickness) in relation to its appropriateness for rodents. Additionally, it should clarify if the implantation device has a flexible characteristic.
5. Describe the fixation method for the ultrasound transducer on the rat's brain, including any medium between the transducer and the brain. Ensure the efficient delivery of acoustic energy to the ultrasound implants.

6. The paper contains grammatical errors and typos. For example, "Deep brain stimulation for epilepsy have" should be "Deep brain stimulation for epilepsy has." Additionally, "charging power up to 2286.62 μw " should be "charging power up to 2286.62 μW ."

Reviewer #3 (Remarks to the Author):

In this manuscript, the authors designed a piezoelectric harvester based on lead-free materials and integrated it in an ultrasound-powered neurostimulator. They demonstrate harvesters operating at two frequencies can be implemented and use this feature to generate a biphasic stimulation pulses for deep brain stimulation in a rat epilepsy model.

The development of lead-free harvesters that rival conventional lead-based devices in performance is a potentially interesting advance. However, the manuscript's focus on epilepsy therapy and brain implants more broadly is less compelling. I recommend that the manuscript be refined to emphasize the key material advancements, rather than placing undue emphasis on epilepsy therapy.

1. The manuscript's emphasis on epilepsy treatment raises significant questions. What is the rationale for ultrasound over clinically-established radio-frequency wireless power transfer, given that the skull is a major barrier and the device shown here does not provide major advantages in miniaturization? What potential benefits does the device offer over clinically approved devices that cannot be achieved with more established technology and warrants the use of new materials (which presents considerable regulatory hurdles)? Furthermore, whether the wireless power results achieved in rodents and a water tank will generalize to humans is unclear. In view of these questions, I am not able to recommend the manuscript in its present form for publication in this journal, but a suitably revised manuscript focusing on the materials aspects with epilepsy treatment as a demonstrative application may be suitable.

2. Simple analog circuits provide a simpler and more reliable way to generate biphasic stimulation pulses, using a smaller footprint and without needing multiple power transfer channels. What are the advantages, if any, of generating biphasic stimulation pulses in this way? How can the pulse be charge balanced if the channels have very different transfer efficiencies? I understand that this functionality can be considered an interesting demonstrative example of dual-frequency harvesters, but this needs to be clear and other potential use cases highlighted, such as having a diversity of harvesters for improved reliability and communication with higher data throughput.

3. The power transfer efficiency as well as the end-to-end efficiency of the system should be quantified and reported (the rectified voltage is not sufficient).

4. What is the sensitivity of the harvester to the orientation relative to the external transducer? Does the design presented here offer any advantages in this respect?

5. For the biocompatibility studies in rodents, where were the devices implanted and are there histological analyses from tissues adjacent to the device? Also, the main text states that the open field test was done 30 days after implantation, whereas Fig. 5d seems to show 10 days.

6. Do the materials investigated here have any potential for use in power transmission, which still relies on lead-based PZT materials?

Dear Editor and Reviewers:

Thank you for your letter and for the reviewers' comments concerning our manuscript entitled "Wireless, biphasic deep brain stimulator for epilepsy based on lead-free dual-frequency ultrasound implants" (Research Article, NCOMMS-23-49482). These comments and suggestions are all valuable and very helpful for revising and improving our manuscript. We have carefully studied the comments, performed supplementary experiments and simulations, given more discussion and explanation, added relevant references, and tried our best to correct and re-edit the main manuscript and the supplementary information that we hope to meet with approval. We have given a more detailed explanation about the significance, impact and usefulness of the device in the revised manuscript. In particular, the supplementary demonstration that US alone does not trigger significant neural activity was also shown in animal experiments. The main modifications are highlighted in the revised manuscript, and the point-to-point responses to the reviewers' comments are listed as follows:

Point-by-point response to the reviewers' comments on MS ID NCOMMS-23-49482

To reviewer #2 (Remarks to the Author):

This paper introduces a wireless and biphasic deep brain stimulator for epilepsy suppression. While the use of ultrasound for energy transfer in electronic stimulation has been reported in several papers, the novelty in this paper lies in the utilization of two ultrasound frequencies to generate biphasic stimulus. However, the advantages of biphasic stimulation need more comprehensive demonstration, and additional experiments should be conducted to compare the results with single-frequency stimulation.

Authors' response: We thank the reviewer very much for carefully assessing our work and giving lots of constructive comments and suggestions needed to improve our

manuscript.

In the revised manuscript, we added literatures as well as additional experiments to elucidate the advantages of biphasic stimulation. Meanwhile, additional control experiments have been conducted to compare the results with single-frequency stimulation. **As shown in Fig. 4d-i**, the antiepileptic effect in the group using single-frequency stimulation (termed as US group) is not significant when compared with the group with stimulation generated by f-BUI under US excitation (termed as US+ f-BUI group).

All issues that address the reviewer's concerns are listed below.

Major comments:

1. The rationale for using two ultrasound frequencies and the advantages of biphasic stimulation in epilepsy inhibition need further clarification. It is advisable to conduct experiments to verify the benefits of biphasic stimulation.

Authors' response: We are grateful to the reviewer for the constructive suggestion.

(1) The rationale for using two ultrasound frequencies is described as following: To produce effective biphasic stimulation within the therapeutic window (100-200 Hz), two SP-1-3 PUEHs with different resonance frequencies (1 MHz and 3 MHz) were deployed and connected them inversely to a pair of rectifier modules, respectively, and eventually to the same stimulation electrode. Transistors prevent currents generated by one SP-1-3 PUEH from propagating through the circuit attached to the other PUEH, ensuring that only one half of the circuit is in operation at a time. As long as the trigger 1 (frequency: 1 MHz, pulse length: 400 μ s, pulse period: 100 Hz) and trigger 2 (frequency: 3 MHz, pulse length: 400 μ s, pulse period: 100 Hz, pulse delay: 200 μ s) are turned on simultaneously at their resonance frequencies, the biphasic pulse can be noticeably observed, whereas the pulse amplitude attenuates significantly once they are switched to the off-resonance frequencies. In addition, the amplitudes of biphasic stimulation waveform can be modulated by adjusting the trigger signals of both channels simultaneously.

(2) For the advantages of biphasic stimulation, previous studies have confirmed that the biphasic waveform ensures that electrochemical reactions that occur in the first phase are reversed in the second phase, resulting in a net zero charge injection into tissue for safety, which allows charge balancing of the electrode, improving stimulation responses and minimizing tissue damage and corrosion of the electrodes during chronic stimulation (*Microsystems & Nanoengineering* 2021, **7**, 62; *Journal of Neuroscience Methods* 2005, **141**, 171-198; *IEEE Transactions on Biomedical Engineering* 2001, **48**, 1065-1070). An increasing number of studies have tended to favor biphasic electrical stimulation over monophasic electrical stimulation (*Nature Biomedical Engineering* 2022, **6**, 706-716; *Nature Communications* 2022, **13**, 7805). At present, most clinically approved electrical neurostimulation therapies utilize biphasic stimulus pulses with “charge balance” to passively or actively charge and discharge the electrodes during each cycle. These advantages are applicable not only to the treatment of epilepsy, but also to electrical stimulation therapy for other neurodegenerative diseases.

Furthermore, to verify the benefits of biphasic stimulation, we conducted additional electrolysis experiments *ex vivo* (Supplementary Fig. 30) and stimulation experiments *in vivo* (Supplementary Fig. 31) using a single US channel (output monophasic pulses: 1 ms, 100 Hz, 1.5 V) and a dual US channel (output biphasic pulses: 1 ms, 100 Hz, +/-1.5 V), respectively. In electrolysis experiments, where the stimulating electrode was immersed in phosphate buffer saline (PBS), bubbles were observed to be generated on the electrodes during single US operation (**Supplementary Fig. 30b**), indicating that water is being electrolyzed and it is harmful. However, no significant electrolysis was found during dual US operation (**Supplementary Fig. 30c**). For stimulation experiments *in vivo*, the duration of monophasic and biphasic stimulation was 30 min per day. After 5 days stimulation, we performed histological analyses of the stimulated tissues. In **Supplementary Fig. 31a,c**, compared to biphasic stimulation, monophasic stimulation increased the extent of the lesion and area of damage, in which monophasic pulses induced a cumulative charge density that created an electrolytic lesion and, consequently, could induce more serious brain tissue damage. Meanwhile, immunofluorescence staining and quantitative

analysis of TNF- α , CD68, and IL-6 were performed to further investigate the inflammatory response caused by brain damage, demonstrating that the expression of inflammatory markers in the biphasic stimulation group was significantly lower than that in the monophasic stimulation group (Supplementary Fig. 31b, d). These results indicate that biphasic stimulation can substantially minimize the inflammatory response and tissue damage, while still providing effective electrical stimulation.

Supplementary Fig. 30 | Comparison of electrolysis experiments by using monophasic pulses and biphasic pulses. (a) Schematic diagram of electrolysis experiment of f-BUI device. (b) Electrode under the microscope during biphasic operation (10 s). (c) Electrode under the microscope during monophasic operation (10 s). Electrolysis occurs on an electrode in PBS solution, as evidenced by gas bubbles. Scale bars are 200 μm .

Supplementary Fig. 31 | Comparison of stimulation experiments *in vivo* by using monophasic pulses and biphasic pulses. (a) Representative H&E staining of rat brain by monophasic and biphasic stimulation. Scale bar is 250 μ m. (b) Representative immunofluorescence staining of inflammatory markers (TNF α , CD68, and IL-6) of stimulation site. Scale bar is 100 μ m. (c) Quantitative analysis of area of damage resulting from different external electrical stimulation ($n = 3$). (d) Quantitative mean fluorescence intensity of TNF α , CD68, and IL-6 in the stimulation site ($n = 3$).

2. The paper states that ultrasound energy is converted to electronic stimulation, but it does not sufficiently address the possibility of ultrasound alone inducing neural activity and epilepsy inhibition. Additional experiments or discussions are needed to distinguish the effects of electronic stimulation from ultrasound.

Authors' response: Thank you for your constructive suggestion. In our revised manuscript, we performed controlled experiments (four groups: the sham group, the seizure group, the US alone (US) group, and the US-induced f-BUI (US+f-BUI) group), as shown in **Fig. 4g-i** in the revised manuscript, in which the ultrasound alone (termed

as “US” group) on epilepsy inhibition is less effective compared to US-induced f-BUI system (termed as “US+ f-BUI” group) according to the results of ECoG, Racine score and survival rate. In particular, the architecture of the CA3 hippocampal regions in the US+f-BUI group was resembling to that in the sham group (**Fig. 4j**). The shrinkage of hippocampal cells in the seizure and US groups increased due to the failure to suppress epilepsy in time, which demonstrated that the epilepsy inhibition of US+f-BUI group is superior to ultrasonic stimulation alone.

Overall, the ultrasound stimulation alone inducing neural activity and epilepsy inhibition is possible but less effective than electrical stimulation. For example, on the one hand, previous reports have demonstrated that the focused ultrasound stimulation can be applied to modulate neural activity at the aimed target of brain due to high spatial resolution and deep focal depth (*Research*, 2019, **2019**, 1748489, 13; *Nature Reviews Neurology*, 2021, **17**, 7-22; *Nature Biomedical Engineering*, 2023, **7**, 149-163). However, we utilized the plane wave ultrasound generated by dual transducer, which failed to modulate neural activity due to lower spatial resolution compared to focused ultrasound stimulation. On the other hand, from the CT image (**Fig. 4c** and **Supplementary Fig. 28**), the f-BUI device was implanted between the skull and skin, near rat’s neck, which made it rare for ultrasound to reach the brain region, which avoided the effect from ultrasound itself on neuromodulation as much as possible. Considering the influence of ultrasound on neuromodulation, we believe that ultrasound (but the sound field needs to be optimized, e.g., focused sound field) and electrical stimulation are expected to achieve synergistic neuromodulation and epilepsy inhibition in future.

In conclusion, we have added more detailed discussion in the revised manuscript to make our opinion clearly. (see lines 313-323, page 12)

3. The acoustic field of the ultrasound is crucial for energy exchange efficiency, and this aspect should not be overlooked. The paper should include the characterization and optimization of the acoustic field to improve energy transfer efficiency.

Authors’ response: We agree with the valuable comment of the reviewer. For

implantable device applications, the acoustic field optimization needs to place primary emphasis on the challenges of transmitter-receiver alignment, which has a significant impact on energy transfer efficiency, especially for dual-frequency operation in our work.

In general, due to the focused architecture, the acoustic energy generated by the focusing transducer is focused into a small area through the confined wave beam, which can improve the magnitude of the acoustic excitation and also reduce energy dissipation caused by the divergence of the sound beam, thereby increasing power conversion efficiency (*Energy & Environmental Science* 2021, **14**, 1490; *Science Advances* 2021, **7**, eabg2507). However, in that focused acoustic field, the focusing point is usually small (usually only millimeters or sub-millimeters for MHz ultrasound), and the energy rapidly decays once it moves away from the focus. As in the FEA simulations, the piezoelectric output of the harvester decreases substantially when it is slightly out of focus (**Supplementary Fig. 2a,c**). This poses a challenge for the alignment of implanted devices, especially the dual-frequency operating system in this work. A slight misalignment will result in a huge difference in efficiency from channel to channel. In addition, focused ultrasound has been shown to have a therapeutic effect (*Nature Reviews Neurology* 2021, **17**, 7–22; *Nature Communications* 2022, **13**, 493) and tends to confuse the electrical stimulation response by causing additional interference with our electrical stimulation devices.

However, in the case of the planar transmitting transducer, the acoustic energy generated is distributed more homogeneously at its front end. Even if the position of the harvester is slightly off (off-center), its acoustically induced piezoelectric potential is not significantly attenuated (**Supplementary Fig. 2b,c**). Thus, the plane configuration has a large positional tolerance, which facilitates the transmitter-receiver alignment of the dual channels in this work.

Supplementary Figure 2 | Comparison of piezoelectric output in the focused and plane-wave acoustic fields when the harvester position is varied. (a1,a2) Schematic showing the change in position of the harvesters in a dual focused acoustic field. **(a3,a4)** Simulated the piezoelectric potential of a harvester as its position changes in a focused acoustic field. **(a3)** At the focus. **(a4)** Not in the focus. **(b1,b2)** Schematic showing the change in position of the harvesters in a dual plane-wave acoustic field. **(b3,b4)** Simulated piezoelectric potential of a harvester as its position changes in a plane-wave acoustic field. **(b3)** At the center. **(b4)** Not in the center. **(c)** Comparison of the piezoelectric output as the harvester position is varied in a focused acoustic field and a plane-wave acoustic field.

In addition, our stimulator is manufactured from a flexible printed circuit board with two receiving piezo-elements at each end that is designed to accommodate the curvature required for implantation above skull. Therefore, the angle of the two elements may not be in a plane when implanted. In general, acoustic receivers usually require special angles related to the ultrasonic beam to achieve optimal performance. The maximum ultrasound intensity will be delivered to the harvester if the acoustic beam is perpendicular to the surface of piezo-elements. Otherwise, the incident ultrasonic power will be weakened if the piezo-elements are tilted at an angle to the acoustic beam. To enhance transfer efficiency and stability and provide design advantages, we redesigned and prepared the external dual-transducer, which connects the two transmitter transducers (1 MHz and 3 MHz) through the use of a soft connection (silicone rubber), as shown in **Supplementary Fig. S1**. As a result, this transmitter adapts to the curved shape of the rat head when attached to the scalp and thus maintains a good angle with the respective receivers (**Supplementary Fig. S1b**), allowing both receivers to have a balanced output. In future research, we will continue to optimize the design of this structure, for example, by using a metamaterial structure to enable multi-angle reception, or by using a flexible two-dimensional array transducer as an external transmitting probe.

Supplementary Fig. 1 | Design and advantages of the external dual-transducer with soft connection. (a) The schematic layout of the dual-transducer. (b) Optical image of the dual-transducer. (c,d) Schematic showing the advantages of a soft connection (c) for the dual-transducer, compared to a rigid connection (d). (e,f) Pictures showing dual-transducers with a rigid connection (e) and a soft connection (f) on a curved surface.

In summary, we have designed a dual plane-wave transmitting transducer using a flexible connection to provide enhanced transfer efficiency and stability. We have also added the above discussion in the revised manuscript to make our opinion clearly.

4. The epilepsy model used in the paper is an acute model, and it is uncertain whether this method is applicable to a chronic model, which is more widely accepted in the epilepsy research community.

Authors' response: We are grateful to the reviewer for the constructive suggestion. In our work, we focused on the design and integration of two newly developed sandwich porous 1-3 (SP-1-3) lead-free piezoelectric components in a flexible printed circuit board, which ultimately developed a flexible, wireless and biphasic deep brain stimulator based on dual-frequency ultrasound implants. On the one hand, the biphasic electrical stimulation is aimed to solve the problem of monophasic and wired electrical stimulation, which allows charge balancing of the electrode, improving stimulation responses and minimizing tissue damage and corrosion of the electrodes during chronic stimulation. On the other hand, as a proof-of-concept, we implanted the device between a rat brain cranium and the skin to investigate the biphasic electrical stimulation for seizure suppression, which demonstrated the feasibility and effectiveness of this device in DBS.

Although a seizure chronic model is more widely accepted in the epilepsy research, the status of seizures in the chronic model is similar to that in the acute model, and many reports have confirmed the electrical stimulation can reduce seizure severity by acute model (*Nature Communications* 2022, **13**, 7805; *Brain Research* 2014, **1593**, 117–125; *Brain* 2019, **142**, 3045–3058; *Life Sciences* 2021, **285**, 119972). Our work is aimed to design a biphasic deep brain stimulator and investigate its feasibility and effectiveness in the suppression of the status of seizures. Meanwhile, the good long-term biocompatibility results demonstrated this device is promising for the research of chronic seizure, other neurodegenerative disease and tissue injury, such as Parkinson's disease, peripheral nerve stimulation, and spinal cord stimulation. We are very appreciate for your constructive suggestion. We will perform more detailed experiments with more models to deeply investigate the mechanism of inhibiting seizure by this device in the future.

Minor comments:

1. The abstract requires improvement. Important information is missing, and the novelty of the paper should be more clearly demonstrated.

Authors' response: We are thankful for the reviewer's valuable suggestion. In the revised manuscript, we improved the abstract by adding important information and emphasizing the novelty of this work. (please see the revised **Abstract**)

2. The introduction should emphasize the advances in epileptic inhibition using deep brain stimulation (DBS) and should reference key literature in the field.

Authors' response: We are grateful to the reviewer for the constructive suggestion. We carefully reviewed the relevant literature. DBS is a powerful tool that can be used to treat brain diseases and investigate their underlying pathophysiology. DBS for the treatment of seizures was approved by the U.S. Food and Drug Administration (FDA) in 2018 (*Epilepsy & Behavior*, 2018, **88S**, 21-24). Three original DBS system manufacturers, Medtronic, Abbott, and Boston Scientific, have developed DBS devices that have been applied in bilateral anterior thalamic nucleus stimulation as adjunctive therapy to reduce the frequency of seizures in individuals with refractory epilepsy (*Epilepsia*, 2010, 51(5), 899-908; *Nature Reviews Neurology*, 2019, **15**, 148–160; *Frontiers in Neurology*, 2022, **13**, 825178). However, technical and clinical challenges also exist. Technical innovation needs to focus on the improvement of practicability, including extension of battery life, design of smaller devices and development of more tailored and adaptive stimulation in addition to the integration of wireless technology, which can alleviate tissue hemorrhage, secondary injury, and massive financial burdens to patients.

In the revised manuscript, we have re-written the **Introduction** section with an emphasis on the advances in epileptic inhibition using DBS, while citing several important references in the field. (see lines 36-47, Page 3)

3. Please provide the full name and description of SP-1-3 PUEHs when they are first mentioned in the main text.

Authors' response: We appreciate the reviewer's suggestion. In the revised manuscript,

we have provided the full name and description of the SP-1-3 PUEHs when they are first mentioned in the main text, which are highlighted in line 132, page 6 and excerpted as follows:

“To achieve this, we employed two sandwich porous 1-3 composite-based piezoelectric ultrasonic energy harvesters (SP-1-3 PUEHs) with different resonant frequencies...” (see line 133, page 6)

4. The paper should address the size of the implantable stimulator (7.3 mm width, 13.6 mm length, 2 mm thickness) in relation to its appropriateness for rodents. Additionally, it should clarify if the implantation device has a flexible characteristic.

Authors' response: We sincerely thank the reviewer for the valuable comments. The size of the flexible f-BUI device is highly suitable for rodents and is similar with the previous reports (*Science Advances* 2022, **8**, eabk0159; *Nature Communications* 2021, **12**, 535; *Proceedings of the National Academy of Sciences of the United States of America* 2021, **118**(30), e2025775118). For example, Jae-Woong Jeong *et al.* developed a soft subdermal implantable device capable of wireless battery charging and programmable controls for applications in deep brain optogenetics. This device is considered compact and lightweight (1.4 g; 19 mm-long × 12 mm-wide × 5 mm-thick) (**Fig. R1**), which allows seamless integration of the device inside the body of small animals such as rats and allows their undisturbed naturalistic behavior and movement (*Nature Communications* 2021, **12**, 535). While our f-BUI device is more flexible, compact and lightweight (7.3-mm width, 13.6-mm length, 2-mm thickness and 0.45-g weight), which does not affect the spontaneous activities of rodents after implantation (**Supplementary video 1**). As a comparison, we have included a table (**Supplementary Table 1**) showing the dimensions of our f-BUI compared to previously reported devices for brain implantation in rodent models. Meanwhile, the f-BUI device was implanted between a rat brain cranium and the skin, suffering multiple bending and pressing deformations induced by the dynamic motions of the rat. After implantation of 30 days, the device is still soft tissue-anchored to the neck region without any displacement (**Supplementary Fig. 28**). Furthermore, we performed experiments using a phantom

skull to study the feasibility of device implantation in the human brain, which demonstrated the f-BUI device is smaller enough than current DBS and is an attractive option for potential applications to the human brain for treatment of neurological or psychiatric disorders (**Supplementary Fig. 29**). In the future, we will further improve the integration and miniaturize the device.

Fig. R1 | The surgery process of soft subdermal implantable device with size of 19 mm-long × 12 mm-wide × 5 mm-thick (*Nature Communications* 2021, 12, 535).

Supplementary Tab. 1 | Comparison of device dimensions for brain implantation in rodent models.

Device	Functionality	Dimensions (long -wide -thick)	Weight	Animal model	Reference
Soft optoelectronic system	Deep-brain optogenetics	19 mm × 12 mm × 5 mm	1.4 g	Rat	(4)
Flexible PUEH device	DBS for analgesia applications	13.5 mm × 9.6 mm × 2.1 mm	0.78 g	Rat	(5)
Subdermally implantable platform	Deep-brain optogenetics	13.5 mm × 10.26 mm × 0.89 mm	87 mg	Mice	(6)
Our f-BUI device	DBS for epilepsy	13.6 mm × 7.3 mm × 2 mm	0.45 g	Rat	This work

Supplementary Fig. 28 | Micro-CT imaging and photographs of f-BUI in the rat brain. (a) Coronal, (b) sagittal orientation and (c) axial orientation. (d) 3D rendering image of the electrode in the rat brain. (e) Photo of rat skin under which the f-BUI device is implanted. (f) and (g) Photos of the rat muscle tissue on which the f-BUI device is implanted. (h) Photo of the implanted f-BUI device after its extraction from the rat after 30 days.

Supplementary Fig. 29 | Front view and top view of skull phantom with the top f-BUI device.

In the revised manuscript, we have modified the text to highlight the flexibility and implantation feasibility of our device. (see lines 139-143, page 6)

5. Describe the fixation method for the ultrasound transducer on the rat's brain, including any medium between the transducer and the brain. Ensure the efficient delivery of acoustic energy to the ultrasound implants.

Authors' response: We thank the reviewers for the valuable suggestion. First, the f-BUI device was subdermally implanted between a rat skull and the skin and fixed with cyanoacrylate glue. Rats were given 10 days to recover after surgery before performing further experiments. Then, the ultrasound stimulation was applied after removing the rat's hair and spreading medical ultrasonic gel on the skin. The dual transducer was aimed at the f-BUI device (based on the micro-CT imaging) and was fixed tightly with a bandage, which formed a head-mounted structure. Medical ultrasonic gel was utilized as a coupling medium for transmitting ultrasound waves between the dual transducer and skin, which made the acoustic energy generated by dual transducer can be effectively transmitted to f-BUI device through the skin with negligible attenuation. We have added details on lines 301-303, page 12 and lines 513-517, page 18:

Lines 301-303, page 12: First, the device was subdermally implanted in a rat brain through the skin, according to the standard procedure (**Fig. 4b, Supplementary Fig. 27** and surgical procedure is detailed in **Methods**).

Lines 513-517, page 18: Specifically, the dual transducer was aimed at the f-BUI device (based on the micro-CT imaging) after removing the rat's hair and spreading medical ultrasonic gel on the skin and was fixed tightly with a bandage, which formed a head-mounted structure. Medical ultrasonic gel was utilized as a coupling medium for efficient delivery of ultrasound waves between the dual transducer and skin.

6. The paper contains grammatical errors and typos. For example, "Deep brain stimulation for epilepsy have" should be "Deep brain stimulation for epilepsy has." Additionally, "charging power up to 2286.62 μw " should be "charging power up to 2286.62 μW ."

Authors' response: We are grateful that the reviewer has pointed out these errors and typos. We have double checked the manuscript and have revised these issues. The modifications are highlighted in the revised manuscript and are listed below:

- “Deep brain stimulation for epilepsy have” → “Deep brain stimulation for epilepsy holds” (line 15, page 2).
- “charging power up to 2286.62 μw ” → “charging power up to 2286.62 μW ” (line 107, page 5).
- “f-BIU system” → “f-BUI system” (line 295, page 11).

To reviewer #3 (Remarks to the Author):

In this manuscript, the authors designed a piezoelectric harvester based on lead-free materials and integrated it in an ultrasound-powered neurostimulator. They demonstrate harvesters operating at two frequencies can be implemented and use this feature to generate a biphasic stimulation pulses for deep brain stimulation in a rat epilepsy model.

The development of lead-free harvesters that rival conventional lead-based devices in performance is a potentially interesting advance. However, the manuscript’s focus on epilepsy therapy and brain implants more broadly is less compelling. I recommend that the manuscript be refined to emphasize the key material advancements, rather than placing undue emphasis on epilepsy therapy.

Authors’ response: We highly appreciate the constructive comments and suggestions of this reviewer on our article. To emphasize the key material advancements, we have re-edited the manuscript. All other issues that address the reviewer’s concerns are listed below.

1. The manuscript’s emphasis on epilepsy treatment raises significant questions. What is the rationale for ultrasound over clinically-established radio-frequency wireless power transfer, given that the skull is a major barrier and the device shown here does not provide major advantages in miniaturization? What potential benefits does the device offer over clinically approved devices that cannot be achieved with more established technology and warrants the use of new materials (which presents

considerable regulatory hurdles)? Furthermore, whether the wireless power results achieved in rodents and a water tank will generalize to humans is unclear. In view of these questions, I am not able to recommend the manuscript in its present form for publication in this journal, but a suitably revised manuscript focusing on the materials aspects with epilepsy treatment as a demonstrative application may be suitable.

Authors' response: We are thankful for the reviewer's valuable suggestion. This work developed a flexible, wireless and biphasic deep brain stimulator for epilepsy treatment based on dual-frequency ultrasound implants, but the innovation focused on the construction of enhanced piezoelectric components as well as on the design of the biphasic stimulator. First, the sandwich porous 1-3 (SP-1-3) lead-free piezoelectric components with enhanced electrical performance was constructed based on multilevel structural engineering, which can generate higher electrical output under ultrasound stimulation. Then, considering the importance of biphasic stimulation, two SP-1-3 PUEHs with different resonance frequencies (1 MHz and 3 MHz) were deployed and connected them inversely to a pair of rectifier modules, respectively, and eventually to the same stimulation electrode, leading to the development of a biphasic deep brain stimulator and to the validation of its potential in biological applications by epileptic animal model.

Naturally, there are definitely some advantages for electrical stimulation generated by ultrasound stimulation. Ultrasound has long been used as a safe and noninvasive tool to diagnose, sense, and monitor diverse diseases and physical conditions. Compared with radio-frequency wireless power transfer, ultrasound could allow wireless power and data transmission through acoustic waves with shorter wavelengths (e.g., millimeter and submillimeter (~ 0.15 - 1.5 mm) at 1-10 MHz ultrasound frequency) (*Science Advances* 2021, **7**, eabf6312; *Nature Biotechnology* 2021, **39**, 855-864). For example, 2-MHz ultrasound waves possess a wavelength of ~ 0.75 mm, while the wavelength of 2-GHz radio-frequency waves is ~ 25 mm. The shorter wavelength enables efficient coupling with tiny electronics, especially miniaturized distributed receivers or array receivers. Meanwhile, ultrasound also allows a higher U.S. Food and Drug Administration (FDA) regulatory limit on power flux

density compared with radio-frequency waves (720 mW cm^{-2} versus 10 mW cm^{-2}), which demonstrated modulation of electrical signals triggered by ultrasound is a safe and promising method as a novel strategy. (Guidance for Industry and FDA Staff (2008)).

As for the effect of skull, the f-BUI device was subdermally implanted between a rat skull and the skin (under the head skin and above the skull). During the procedure, the skull was exposed and the burr holes were drilled at the coordinates to implant stimulating electrodes, and the wound was sutured the after surgery. Thus, the ultrasound is directed to the device through the skin tissue and is not affected by the skull. In the experiment, the ultrasound stimulation was applied after removing the rat's hair and spreading medical ultrasonic gel on the skin. Medical ultrasonic gel was utilized as a coupling medium for transmitting ultrasound waves between the dual transducer and skin, which made the acoustic energy generated by dual transducer can be effectively transmitted to f-BUI device through the skin with negligible attenuation. To validate the effectiveness of ultrasound subcutaneous delivery, we measured the output of the f-BUI device at different tissue thicknesses and verified the results (**Fig. 4g**). Therefore, here, the skull is not a major barrier for the subcutaneous transmitting of ultrasound stimulation and won't affect the generation of ultrasound-triggered electrical signals.

The f-BUI system highlights its lightweight electrical design and flexible, soft mechanical features, with overall dimensions of 7.3 mm in width, 13.6 mm in length, and 2 mm in thickness, which is small size and can generate voltage output comparable to the clinically approved devices. Meanwhile, the f-BUI system can achieve wireless-powered battery-free neuromodulation driven remotely by programmable ultrasound pulses, which does not require costly chip design and is superior to current clinical battery-powered bulky neurostimulators. Given the advantages of the device, our work is aimed to investigate the feasibility and effectiveness of this device for bipolar deep brain radio stimulation in epilepsy, which demonstrated this device is able to suppress seizure by ultrasound-triggered bipolar electrical stimulation and is promising for the research of other neurodegenerative disease and tissue injury, such as Parkinson's

disease, peripheral nerve stimulation, and spinal cord stimulation. However, this device needs further optimization and detailed experiments in large animals and humans are necessary to eliminated potential risks in the future for eventual clinical translation.

Following the reviewer's comments and suggestions, the above discussion has been added to the main text with highlighting, and we have re-edited the manuscript to focus more on the materials aspects with epilepsy treatment as a demonstrative application.

2. Simple analog circuits provide a simpler and more reliable way to generate biphasic stimulation pulses, using a smaller footprint and without needing multiple power transfer channels. What are the advantages, if any, of generating biphasic stimulation pulses in this way? How can the pulse be charge balanced if the channels have very different transfer efficiencies? I understand that this functionality can be considered an interesting demonstrative example of dual-frequency harvesters, but this needs to be clear and other potential use cases highlighted, such as having a diversity of harvesters for improved reliability and communication with higher data throughput.

Authors' response: We agree with the valuable comment of the reviewer.

(1) In our manuscript, the advantages of this proposed scheme using two ultrasonic channels for generating biphasic stimulation pulses can be summarized in the following 2 points: **First**, the device features a simplistic layout and a compact design that occupies a smaller footprint, eliminating the need for complex arithmetic circuits. Generally, in existent device designs, complex arithmetic circuits or even application specific integrated circuits (ASICs) are required to generate biphasic stimulation waveforms. It is not only expensive to design and manufacture, but also challenging to miniaturize the device size due to the use of multiple components, thus making it unfavorable for implantable applications. Our design incorporates multiple ultrasonic channels in a simplistic layout, where the advantage of short ultrasonic wavelengths (e.g., ~1.5 mm for 1 MHz US) for efficient coupling to miniature receivers was leveraged, thus addressing the above issue to a certain extent. **Second**, the stimulation waveform (e.g., pulse length, amplitude, frequency) can be more easily regulated by

controlling the external transmitter. In a general pre-customized arithmetic circuit, the output signals are usually limited to one or more pre-designed results and can not be externally adjusted to achieve richer waveforms. As a result, our design is highly scalable and only requires corresponding regulation of the external transmitter signals on request, allowing for targeted applications in more cases in the future.

(2) We agree with the reviewer that how to balance the charge of the pulse is indeed a matter of concern. For both channels, although their corresponding transmitters and receivers are of the same material and construction design, the difference in ultrasonic frequency does result in a difference in transmission efficiency, for example, one of the reasons for this is the greater attenuation of 3 MHz ultrasound (**Fig. 3g**). In the experiment, the induced output of 3-MHz ultrasound is indeed smaller than 1-MHz ultrasound for the same trigger signal condition, with an end-to-end efficiency (for 3 MHz channel) of ~82% of 1-MHz channel, thus resulting in a charge imbalance in the biphasic pulse. To effectively compensate for the imbalance in charge, we increased the trigger 2 signal for the 3-MHz channel by nearly 22% by through experiment tests (**Supplementary Fig. 24**). From the test results (**Fig. 3k**, red line), we can see that by compensating the trigger signal at the input, the output pulse can be relatively balanced, thus providing better assurance for subsequent stimulation applications.

Supplementary Fig. 24 | Dual-alternating current (AC) trigger waveforms. (a) Trigger 1 and trigger 2 waveforms without compensation. (b) Trigger 1 and trigger 2 waveforms with compensation. The inset shows an enlargement of the trigger waveforms. Scale bars, 0.02 V and 500 μ s. (c,d) Enlargement of the trigger waveforms in (b).

Fig. 3 | (k) Biphasic stimulus pulse generated by the f-BUI when both trigger signals are turned on simultaneously. The charge imbalance due to different channel

transmission efficiencies can be adjusted by compensating for the 3-MHz trigger signal. The inset shows the enlargement of biphasic waveforms. Scale bars are 0.5 ms and 0.5 V.

(3) We thank the reviewers for the valuable suggestion. At present, all clinically approved electrical neural stimulation therapies use various forms of “charge balanced” biphasic stimulation waveforms that actively or passively charges and discharges the electrode with each cycle. In the wireless implantable device, we have achieved adjustable biphasic stimulation waveforms by using dual-frequency ultrasound channels, which has advantages over previous ultrasound delivery of monophasic stimulation and does not require complex arithmetic circuits or even application specific integrated circuits (ASICs), providing an attractive design strategy for future stimulation-related therapies.

In addition to the biphasic stimulation functionality demonstrated in this study, such device layout can also be scaled to more ultrasonic frequency channels (more than two) for increased reliability and communication capabilities with higher data throughput. Since each channel can be independently manipulated, it is possible to implement arrayed layouts for zoned stimulation of brain tissue, without interfering with each other. Furthermore, it is possible to realize multiple functionalities in a single integrated device, such as one ultrasound channel for information transmission and one for power transmission, thus realizing real-time monitoring and treatment of the human body.

Following reviewer’s comments and suggestions, the above description has been added to the main text in the revised manuscript with highlighting. (see lines 83-91, page 5, lines 279-293, page 11 and lines 406-408, page 15)

3. The power transfer efficiency as well as the end-to-end efficiency of the system should be quantified and reported (the rectified voltage is not sufficient).

Authors’ response: We sincerely thank the reviewer for the valuable comments. To evaluate the power transfer efficiency of our ultrasound system, the device was characterized in a water tank at a 15-mm distance (**Fig. 3a**). Water has an acoustic

impedance similar to that of soft tissue ($\sim 1.5 \text{ MRayl}$).

Since our system is operated by two ultrasound frequencies, we characterized the 1-MHz and 3-MHz channels, respectively. For the 1-MHz channel (the resonant frequency of both the transmitting transducer and the energy harvesting element is 1 MHz), the system exhibited a power transfer efficiency from the acoustic power (617 mW) at the implant's piezo surface to the implant's electrical output power (156 mW) of $\sim 25.3\%$ and an end-to-end efficiency of $\sim 7.8\%$, defined as the ratio of the implant's electrical output power (156 mW) to the consumed power (2 W) of the external transmitting transducer. For the 3-MHz channel, the system exhibited a power transfer efficiency from the acoustic power (578 mW) at the implant's piezo surface to the implant's electrical output power (130 mW) of $\sim 22.5\%$ and an end-to-end efficiency of $\sim 6.5\%$, which is the ratio of the implant's electrical output power (130 mW) to the consumed power (2 W) of the external transmitting transducer.

The acoustic power at the implant's piezo surface was calculated via the acoustic field intensity data from the hydrophone over the surface of the implant piezo. The implant's electrical output power was measured the instantaneous power at load resistance of 150 ohms. The total transmission efficiency depends on factors such as the electro-acoustic conversion of the external transducer, frequency-dependent ultrasonic attenuation in the propagation medium, reflections, acoustic-electric conversion of the implant, and impedance matching with the circuit. Future research could start with these aspects and continue to improve the transmission efficiency of the system. Here, the 3-MHz channel presents a lower efficiency than 1-MHz one because the higher the frequency the faster it attenuates in the propagation media. Although the two channels present different transmission efficiencies, we can achieve a balance between the two channels by adjusting the input power of the external transmitting transducer.

Following reviewer comments, the above description has been added to the main text in the revised manuscript with highlighting. (see lines 243-253, page 10)

4. What is the sensitivity of the harvester to the orientation relative to the external

transducer? Does the design presented here offer any advantages in this respect?

Authors' response: We thank the reviewer for the constructive reviewing. In general, acoustic receivers usually require special angles related to the ultrasonic beam to achieve optimal performance. The maximum ultrasound intensity will be delivered to the harvester if the acoustic beam is perpendicular to the surface of piezo-elements. Otherwise, the incident ultrasonic power will be weakened if the piezo-elements are tilted at an angle to the acoustic beam (*Science Advances*, 2022, **8**, eabk0159; *Energy & Environmental Science*, 2021, **14**, 1490-1505). To study the sensitivity of the harvester to the orientation relative to the external transducer, we simulated and measured the change in the output of the harvester in response to the change in its angle relative to the external transducer, as shown in the **Supplementary Fig. 21**. According to the simulated and measured results, as the incidence angle becomes larger, the output drops. The harvester had a wide acceptance angle, with piezo-voltage harvested varying between 58% and 100% of maximum over a 30° angular misalignment range of the harvester relative to the acoustic field. We have added the results to the main text in the revised manuscript with highlighting. (see lines 261-266, page 10)

Supplementary Figure 21 | Relationship between the output of the harvester and the incidence angle of the ultrasound wave. (a) Schematic showing the experimental test. (b) Simulated piezo-potential of a SP-1-3 piezo-harvester under ultrasonic excitation when the harvester is parallel to the transmitter (b1) and rotated by 30° (b2). (c) Measured output voltage signal of the harvester at different US incident angles.

In addition, our harvester is a flexible device, which adapts to the curved structure of the skull during implantation, with two receiving piezo-elements at each end of the flexible printed circuit boards. Therefore, the angle of the two elements may not be in a plane when implanted. To enhance transfer efficiency and stability and provide design advantages, we redesigned and prepared the external dual-transducer, which connects the two transmitter transducers (1 MHz and 3 MHz) through the use of a soft connection (silicone rubber), as shown in **Supplementary Fig. S1**. As a result, this transmitter adapts to the curved shape of the rat head when attached to the scalp and thus maintains

a good angle with the respective receivers (**Supplementary Fig. S1b**), allowing both receivers to have a balanced output. In future research, we will continue to optimize the design of this structure, for example, by using a metamaterial structure to enable multi-angle reception, or by using a flexible two-dimensional array transducer as an external transmitting probe.

Supplementary Fig. 1 | Design and advantages of the external dual-transducer with soft connection. (c,d) Schematic showing the advantages of a soft connection (c) for the dual-transducer, compared to a rigid connection (d). (e,f) Pictures showing dual-transducers with a rigid connection (e) and a soft connection (f) on a curved surface.

5. For the biocompatibility studies in rodents, where were the devices implanted and are there histological analyses from tissues adjacent to the device? Also, the main text states that the open field test was done 30 days after implantation, whereas Fig. 5d seems to show 10 days.

Authors' response: We thank the reviewer for the constructive reviewing.

(1) In this work, our devices were encapsulated with silicon elastomer membrane

(Ecoflex-0030) and implanted under the head skin and above the skull. During the procedure, the skull was exposed and the burr holes were drilled at the coordinates to implant stimulating electrodes. The wound was sutured after surgery. For the biocompatibility studies in rodents, we supplemented the histological analysis from tissues adjacent to the device, as shown in **Fig. 5g-j**. The results showed that the adjacent tissues did not exhibit obvious inflammatory reaction and surrounding tissue damage after f-BUI implantation. In addition, it has been shown that silicon elastomer membrane (Ecoflex-0030) possesses good biocompatibility and is often used for encapsulation of implanted devices (*Nature* 2019, **565**, 361-365; *Small* 2023, **19**, 2206839).

(2) For **Fig. 5d**, which shows “10 days”, this is a clerical error and should be “30 days”, consistent with the main text. Rats were given 10 days to recover after surgery before performing further experiments, and the open field test was performed on 30 days to investigate the biological safety of the device. We are sorry for our carelessness and thank the reviewer for catching this error. We have corrected this in the **Fig. 5d** in the revised manuscript to avoid the misunderstandings (see **Fig. 5d**).

Following reviewer comments, the additional results of histological analysis have been added to the main text in the revised manuscript with highlighting. (see lines 357-361, page 14)

Fig. 5 | Demonstration of the biological safety of f-BUI. **a** Schematic diagram of examining cytocompatibility of SP-1-3 piezoelectric elements. **b** Live (green)/Dead (red) cells staining treated with/without SP-1-3 piezoelectric element leaching liquors on day 1, 3, and 5. Scale bar, 100 μm . **c** Corresponding cell viability assessed by CCK8 assay. **d** Experimental procedure of f-BUI biological safety in vivo. **e** Representative motion route heatmap in Sham, f-BUI, US, and US+f-BUI group. **f** Total motion distance of SD rats with the different treatment derived from open field test ($n = 5$). **g** H&E staining images of skin, heart, liver, spleen, lung and kidney of the rats with/without the implantation of f-BUI. Scale bars are 200 μm . **h** Quantitative analysis of areolar tissue thicknesses, indicating there are

no significant differences between two groups ($n = 3$). **i** Representative immunofluorescence images of inflammatory response (TNF α , CD68, and IL-6) of implantation site. Scale bar is 50 μm . **j** Quantitative mean fluorescence intensity of TNF α , CD68, and IL-6 in the implantation site ($n = 3$).

6. Do the materials investigated here have any potential for use in power transmission, which still relies on lead-based PZT materials?

Authors' response: We thank the reviewer for the comments and suggestions.

For power transmission, the parameters of greatest interest for piezoelectric materials are the mechanical quality factor (Q_m , inverse of $\tan \delta$), and such high- Q_m materials are mainly performed by hard PZT ceramics, such as PZT-4 and PZT-8 ceramics, which possess both high Q_m (~800-1500) and high d_{33} (~250-350 pC/N). In particular, a high Q_m prevents a significant increase in temperature due to the heat generated by the working device, which therefore determines the transmission stability of the transmission device, especially in high-intensity ultrasound applications, high intensity focused ultrasound (HIFU).

However, in our current study, the implantable piezoelectric system used for energy receiver is the sodium potassium niobate (K,Na)NbO₃-based lead-free piezoelectric materials, which typically feature a high piezoelectric coefficient d_{33} accompanied by a high dielectric constant ϵ , analogous to commercial soft PZT ceramics (e.g., PZT-5). Typically, this type of piezoelectric material is more suitable for piezoelectric sensing, such as used in this work to receive ultrasound energy. Since it is used as a receiving application, to optimize the ultrasound receiving sensitivity, here we further manufactured a sandwich porous 1-3 (SP-1-3) composite structure of multi-layered piezoelectric micro-rods embedded inside an epoxy matrix to improve the harvesting figure of merit ($\text{FoM} = d_{33} \times g_{33}$). Such a strategy focuses on maintaining a high piezoelectric coefficient (d_{33}) and lowering the dielectric constant (ϵ) of the piezoelectric system, thereby enhancing the receiving performance.

Unfortunately, however, the KNN-based piezoelectric systems are difficult to obtain both high Q_m and high d_{33} simultaneously. Generally, outstanding KNN-based ceramics can achieve piezoelectric coefficient up to 350-500 pC/N, but Q_m is only a few dozen ($\sim 50-100$). Even the most advanced high- Q_m KNN-based lead-free systems (e.g., $d_{33} \sim 300$ pC/N, $Q_m \sim 250$) still fall far short of the PZT-4 and PZT-8. Therefore, current KNN-based lead-free piezoelectric ceramics are not suitable for power transmission. Currently, researchers are improving the KNN-based system through doping, component modulation, and process optimization, hoping to catch up with the hard Pb-based series soon.

In addition, air holes have been introduced in our composite structure in order to reduce the dielectric constant, and these air holes increase the damping of the system. If SP-1-3 composite is used as a power transmission, these dampings increase the energy loss capability and the power is dissipated by converting it into thermal energy, which does not result in an effective energy emission. To compare the conventional 1-3 composite with our SP-1-3 composite, we simulated the acoustic emission properties of both materials using COMSOL-based finite element simulation, as shown **Fig. R2** below. The results show that the acoustic power emitted by the SP-1-3 composite with pores is less than that of the conventional 1-3 composite, further confirming our conclusion above.

Fig. R2 | Simulated comparison of acoustic emission from piezo-composites using COMSOL multiphysics. (a) Conventional 1-3 piezo-composite. (b) SP-1-3 piezo-

composite. The acoustic power emitted by the SP-1-3 composite with pores is less than that of the conventional 1-3 composite.

Nonetheless, in ultrasound wireless transmission for medical systems, the transmitting probe is mainly used outside the body and will effectively reduce its toxicity in an intact encapsulation. As for the part used for implantation, we mainly consider lead-free ones to minimize the possible lead hazards. In the future, we will also develop high- Q_m lead-free piezoelectric materials for use in transmitter components outside the body.

The above discussion has been briefly added to the **Discussion** section. (see line 416-419, page 16)

REVIEWER COMMENTS

Reviewer #2 (Remarks to the Author):

Authors included additional experimental results in the revision, this is very well appreciated. However, the inhibitory effect of the biphasic deep brain stimulator should be demonstrated.

Reviewer #3 (Remarks to the Author):

In this revision, the authors provide results and added text to clarify the specific points raised by the reviewers. The new additional experimental and simulation results satisfactorily address my technical concerns. However, the revisions to the main text, in my opinion, move the framing of the work significantly in the wrong direction, to the extent of being misleading.

I request that the authors substantially revise the main text to address the following points.

1. The authors have doubled-down on their emphasis on epilepsy rather than refocusing the contributions on the development of high-performance lead-free harvesters. This is problematic because the authors rely on misleading statements to justify this angle (see next point). I ask that the authors significantly de-emphasize epilepsy (by deleting text) and discuss the materials advances. Here are some specific suggestions:

a. Consider changing the title as "Lead-free dual-frequency ultrasound implants for wireless, biphasic deep brain stimulation"

b. Delete the first sentence of the abstract.

c. Replace "suffer from deficiencies in eco-friendly materials engineering and device integration design" with specific challenges in biocompatibility and harvesting performance related to lead/lead-free materials

d. Replace the last sentence of the abstract to emphasize the materials advance and specific potential applications.

e. Rewrite the first paragraph of the introduction. This should start with DBS and discuss the device challenges. It is misleading to start with epilepsy because DBS is used to treat only a small fraction of these cases - regardless this is not the main advance of this paper.

f. Please carefully go through the main text and delete overly enthusiastic statements like "bringing numerous benefits", "even promoting the advancement of medicine", "great significance for future medical therapies".

2. The authors have incorporated several incorrect or misleading statements in the manuscript as a part of their rebuttal. I ask that the authors delete or correct these. In many cases, they may simply acknowledge the challenges in the discussion section and discuss how they can be addressed in future work. However, they should not be ignored.

a. "US waves offer superior safety and efficiency advantages". Speaking as someone who is enthusiastic about ultrasound wireless power: this is patently false. Both modalities are considered safe under appropriate exposure levels and inductive coupling systems used in clinical devices such as cochlear implants routinely achieve efficiencies >80%. The wavelength/power density is irrelevant when comparing to inductive systems - these involve magnetic fields which does not interact with the body at all. What is correct is that ultrasound is a promising, early stage technology that may offer better performance for small (mm-scale) and deep (several cm) implants.

b. "...portable external ultrasonic sources can provide programmable, adjustable, and long-term sustainable power delivery in vivo". This is also openly misleading. I am not aware of a single clinical example of continuous ultrasound power delivery (they are all used episodically), whereas there are already many RF devices already clinically used. There are substantial challenges such as skin-transducer coupling that are unresolved. Delete this and discuss the challenges at the end of the manuscript.

c. "eco-friendly", "environmental pollution from discarded batteries". Medical devices represent a minuscule fraction of total electronic waste while providing enormous human benefit. Why not discuss the biocompatibility concerns of lead-based materials instead?

d. "the use of ASICs for implants is not only expensive to design and manufacture". This is incorrect. There are many ICs used in medical devices on the market that cost a few dollars or cents. Every clinically-used stimulator that I am aware of uses an IC for generating stimulation pulses. Please delete this statement, discuss the need for ICs as future work, and highlight other use cases, such as "having a diversity of harvesters for improved reliability and communication with higher data throughput".

Reviewer #4 (Remarks to the Author):

Wang et al., present work on a novel deep brain stimulator (DBS) based on lead-free dual-frequency ultrasound implants. The idea is to develop a way to power DBS stimulators wirelessly over long-periods of time.

I am an epilepsy researcher with expertise in pre-clinical in vivo rodent models and I was brought in to review the physiology and seizure data from the animal model. I find that these elements of the paper are not well done and in its current state I do not recommend this article for publication. The quality of data presentation is low, and similarly how the seizures are measured, and how they are tested statistically, does not convince me that this approach is working. See below further comments:

I was surprised by the choice of model. Although acute systemic injection of 4-aminopyridine (4-AP) is used as a model of epilepsy in the literature, it is not very common. Other models that are more commonly used are injections of pentylenetetrazole or kainic acid. The issue with 4-AP is the increased presence of inhibitory GABAergic activity that may not necessarily mimic true human epilepsy. See:

4-Aminopyridine induces a long-lasting depolarizing GABA-ergic potential in human neocortical and hippocampal neurons maintained in vitro M Avoli, P Perreault, A Olivier, JG Villemure - Neuroscience letters, 1988 –

4-aminopyridine-induced epileptiform activity and a GABA-mediated long-lasting depolarization in the rat hippocampus P Perreault, M Avoli - Journal of Neuroscience, 1992 –

Nonetheless, this is really a proof of principle study, so it could be valid to use this model. But the data thus far is not very convincing:

It is difficult for me to understand precisely what is being presented and how the stimulation affects overall brain activity. It would be important to show that a dataset where there is a clear effect of brain stimulation on brain activity. At the moment we only have the indirect effect on seizures. Basically, we would want to see the SHAM + US+f-BUI (non-seizure) group. Is there an effect on the electrographical data? This would also be helpful in dissociating whether the claimed effects on power reduction are specifically on seizures or on overall brain activity. Or just to prove that there is any effect at all of the stimulation. It would be useful to see plots of the electrographical data around the start of stimulation to see the effect on local field potential (LFP) activity. What does this high frequency stimulation do to overall brain activity is my query?

In the zoomed in time scale data across figure 4d of the electrographical data there is no scale bar. The end result showing decreased overall power in the spectrograms in d suggests that there is a decrease in overall power, but not complete return to normality or the conditions shown in sham. A better analysis could be to take shorter time periods. In acute models of epilepsy seizure-like events often come in bursts with breaks in the activity, we see this a bit in plots with long time periods of LFP data, with seizures seemingly waxing and waning. This on-and off type pattern is still seen in the US+f-BUI condition after stimulation, there is still an oscillatory rhythm which is not present in the sham conditions. Could the experimenters quantify instead the number of bursts of events? Or take something like a coastline measure? But, overall, it would be nice to see the immediate effect on seizures, not after stimulation, but as the stimulation comes on, in real time, with a clear demarcation of a before and after stimulation from the same continuous data. What is the transition between high seizure activity to low seizure

activity after 15 minutes of stimulation? 15 minutes is a long time in terms of acute models, so try to break it down into shorter periods.

4e and f do not show a decrease in power. We do not see individual variability in power, there are no stats calculated to show that there are differences between the various groups. Could this be done in different spectral frequency bands? Similarly in 4g I do not understand the stats that are performed to reach the conclusion. You need to be able to show individuals and more than the comparison between the condition before and after the stimulation, you would want to compare across groups with a 2-way ANOVA type statistical tests. But overall I think it is critical to show a non-epilepsy stimulated group. The racine score experiment is okay but we need detailed stats and plots with individual animals shown. Furthermore, it would be great to see a progression of epileptic activity? Perhaps the authors could plot how quickly high racine scores are reached and compare this across groups.

Finally, 4J is not convincing at all. For me the images show different exposition levels. Can the authors quantify the number of cells? Can this be done in other structures in hippocampus and beyond? And, again, we would need a non-epileptic stimulated group to make sure there is no effect on seizures. Is there evidence in the literature that there is hippocampal sclerosis in this model?

One positive element is the survival. In a way this model is both acute and chronic as survival is measure in the model. Repeated measure ANOVAs should be used for the stats here and be well described.

It is not clear to me the mechanism of how the anti-epileptic effect works, we need some incite into the circuitry involved. Why are the electrodes implanted in the locations stated, and what are they. I think recording is in the hippocampus while stimulation is in a more rostral structure, why these locations? How far reaching would the stimulus be beyond the stimulated structure?

Finally, I concur with the other reviewers that the exact value of this approach is not clear as current DBS stimulators may be easier to control and already quite efficient.

Point-by-point response to the reviewers' comments on MS ID NCOMMS-23-49482A

Dear reviewers,

We deeply appreciate the effort that you have taken in reviewing our manuscript entitled “Lead-free dual-frequency ultrasound implants for wireless, biphasic deep brain stimulation” (revised title) (NCOMMS-23-49482A). These comments and suggestions are all valuable and very helpful for revising and improving our manuscript. We have carefully studied the comments, performed supplementary experiments, given more discussion and explanation, and tried our best to correct and re-edit the main manuscript and the supplementary information that we hope to meet with approval.

In particular, we have given a more detailed demonstration about the the inhibitory effect of the biphasic deep brain stimulator. In response to the 4-AP model raised by Reviewer #3 that 4-aminopyridine (4-AP) is used as a model of epilepsy but it is not common and it would be nice to have more common models, we have made a detailed discussion, and we have added the experimental results with a SHAM + US+f-BUI (non-seizure) group.

The main modifications are highlighted in the revised manuscript. Here, we have also included a point-by-point response to the referee's comments (the original comments are in **black** color and our responses are in **light blue**), as shown below:

Reviewer #1 (Remarks to the Author):

Authors included additional experimental results in the revision, this is very well appreciated. However, the inhibitory effect of the biphasic deep brain stimulator should be demonstrated.

Authors' response: We greatly appreciate once again the reviewer for your

constructive suggestions, which have improved our manuscript a lot. For the inhibitory effect of the biphasic deep brain stimulator in antiepileptic therapy, the experimental results can be found as shown in **Figure 4**. In our experiments, there were four groups: the sham group, the seizure group, the US alone (US) group, and the US-induced f-BUI (US+f-BUI) group. The US-induced stimulation was performed by an external dual-frequency transmitter (1 and 3 MHz), which can trigger the f-BUI device to generate biphasic electrical stimulation (output biphasic pulses: 1 ms, 100 Hz, +/-1.5 V) as shown in **Fig. 3k**. As shown in **Fig. 4d**, when the seizures reached stage 5 according to the Racine score, representative electrocorticography (ECoG) signals and related time-frequency spectrum in the seizure, US, and f-BUI groups showed typical sharp epileptic waves with greater amplitudes compared to the sham group. After 15 min of US stimulation (0.2 MPa, 100 Hz pulses), sharp epileptic waves significantly decreased and disappeared in the US+f-BUI group under biphasic electrical stimulation, demonstrating that the antiepileptic effect of this system is more effective than that of other groups. Meanwhile, we have added the 2-way ANOVA type statistical tests to the results of the ECoG power spectral density (PSD) and Racine score (**Fig. 4g&h**), demonstrating the seizure severity in the US+f-BUI group declined dramatically, which indicated that the inhibitory effect of the biphasic deep brain stimulator in antiepileptic therapy is significant. The above discussion has been added and revised and highlighted in the revised manuscript.

Reviewer #2 (Remarks to the Author):

In this revision, the authors provide results and added text to clarify the specific points raised by the reviewers. The new additional experimental and simulation results satisfactorily address my technical concerns. However, the revisions to the main text, in my opinion, move the framing of the work significantly in the wrong direction, to the extent of being misleading.

Authors' response: Again, we highly appreciate the constructive suggestions of this reviewer on our manuscript. These suggestions are highly important and we have followed these guidelines in revising the main text. All revisions are listed below and highlighted in the main text.

I request that the authors substantially revise the main text to address the following points.

1. The authors have doubled-down on their emphasis on epilepsy rather than refocusing the contributions on the development of high-performance lead-free harvesters. This is problematic because the authors rely on misleading statements to justify this angle (see next point). I ask that the authors significantly de-emphasize epilepsy (by deleting text) and discuss the materials advances. Here are some specific suggestions:

a. Consider changing the title as “Lead-free dual-frequency ultrasound implants for wireless, biphasic deep brain stimulation”.

Authors' response: We are thankful for the reviewer's valuable suggestion. The title of the manuscript has been changed as “**Lead-free dual-frequency ultrasound implants for wireless, biphasic deep brain stimulation**”.

b. Delete the first sentence of the abstract.

Authors' response: Thank you for your suggestion. The first sentence of the abstract has been deleted.

c. Replace “suffer from deficiencies in eco-friendly materials engineering and device integration design” with specific challenges in biocompatibility and harvesting performance related to lead/lead-free materials.

Authors’ response: We are grateful to the reviewer for the constructive suggestion. We have replaced this description with “specific challenges in biocompatibility and harvesting performance related to lead/lead-free materials” in the abstract.

d. Replace the last sentence of the abstract to emphasize the materials advance and specific potential applications.

Authors’ response: Thanks for the suggestion. We have replaced the last sentence of the abstract.

e. Rewrite the first paragraph of the introduction. This should start with DBS and discuss the device challenges. It is misleading to start with epilepsy because DBS is used to treat only a small fraction of these cases - regardless this is not the main advance of this paper.

Authors’ response: We thank the reviewer for the valuable suggestion. In the revised manuscript, we have rewritten the first paragraph of the introduction, which starts with DBS as a background and discusses the challenges for current DBS devices. This rewritten section is copied below:

“Deep brain stimulation (DBS) has emerged as an effective treatment strategy for a wide range of neurological conditions, such as Alzheimer’s disease^{1,2}, Parkinson’s disease^{3,4}, essential tremor^{5,6}, and epilepsy⁷⁻⁹. DBS technology involves surgically implanting electrodes into specific areas of the brain that deliver programmable electrical pulses to modulate abnormal brain discharges, thereby ameliorating or treating disease¹⁰⁻¹³. The power supply for DBS, however, remains a fundamental challenge to be addressed. Conventional external power supplies use transcutaneous or percutaneous wires, which are inconvenient, easily damaged, and susceptible to infection, especially in long-term applications¹⁴⁻¹⁶. An alternative is to integrate the

battery with the implants, however, regular surgery is required for patients to replace the dead batteries with new ones due to the limited life of batteries, which will undoubtedly add to the physical pain and financial burden of patients¹⁷⁻¹⁹.”

f. Please carefully go through the main text and delete overly enthusiastic statements like “bringing numerous benefits”, “even promoting the advancement of medicine”, “great significance for future medical therapies”.

Authors’ response: We are thankful for the reviewer’s suggestion. We have gone through the full main text and delete all overly enthusiastic statements.

2. The authors have incorporated several incorrect or misleading statements in the manuscript as a part of their rebuttal. I ask that the authors delete or correct these. In many cases, they may simply acknowledge the challenges in the discussion section and discuss how they can be addressed in future work. However, they should not be ignored.

a. “US waves offer superior safety and efficiency advantages”. Speaking as someone who is enthusiastic about ultrasound wireless power: this is patently false. Both modalities are considered safe under appropriate exposure levels and inductive coupling systems used in clinical devices such as cochlear implants routinely achieve efficiencies >80%. The wavelength/power density is irrelevant when comparing to inductive systems - these involve magnetic fields which does not interact with the body at all. What is correct is that ultrasound is a promising, early stage technology that may offer better performance for small (mm-scale) and deep (several cm) implants.

Authors’ response: We sincerely thank the reviewer for the valuable comment and suggestion. In the revised manuscript, we have revised that description, replacing it with “ultrasound is a early stage but promising technology that may offer better performance for small (mm-scale) and deep (several cm) implants...”.

b. “...portable external ultrasonic sources can provide programmable, adjustable, and long-term sustainable power delivery in vivo”. This is also openly misleading. I am not

aware of a single clinical example of continuous ultrasound power delivery (they are all used episodically), whereas there are already many RF devices already clinically used. There are substantial challenges such as skin-transducer coupling that are unresolved. Delete this and discuss the challenges at the end of the manuscript.

Authors' response: We thank the reviewers for the suggestion. Current ultrasonic power delivery does present challenges as pointed out by the reviewer. We have removed this misleading statement from the main text and discussed these challenges at the end of the manuscript.

c. "eco-friendly", "environmental pollution from discarded batteries". Medical devices represent a minuscule fraction of total electronic waste while providing enormous human benefit. Why not discuss the biocompatibility concerns of lead-based materials instead?

Authors' response: Thanks to the reviewer. We agree with this view. In the revised manuscript, we have removed these descriptions of "eco-friendly", "environmental pollution from discarded batteries", thus focusing the discussion more on the biocompatibility of lead-based piezoelectric materials.

d. "the use of ASICs for implants is not only expensive to design and manufacture". This is incorrect. There are many ICs used in medical devices on the market that cost a few dollars or cents. Every clinically-used stimulator that I am aware of uses an IC for generating stimulation pulses. Please delete this statement, discuss the need for ICs as future work, and highlight other use cases, such as "having a diversity of harvesters for improved reliability and communication with higher data throughput".

Authors' response: We sincerely thank the reviewer for the valuable suggestion. In the revised manuscript, we have removed the misleading statement. We also discussed the need for ICs for future work and highlighted some use cases in the discussion section, which emphasizes the multi-frequency designs having a diversity of harvesters for improved reliability and communication with higher data throughput.

Reviewer #3 (Remarks to the Author):

Wang et al., present work on a novel deep brain stimulator (DBS) based on lead-free dual-frequency ultrasound implants. The idea is to develop a way to power DBS stimulators wirelessly over long-periods of time.

Authors' response: We thank the reviewer very much for carefully assessing our work and giving lots of professional comments and suggestions that will improve our manuscript. The point-to-point responses and discussion are as follows:

I am an epilepsy researcher with expertise in pre-clinical in vivo rodent models and I was brought in to review the physiology and seizure data from the animal model. I find that these elements of the paper are not well done and in its current state I do not recommend this article for publication. The quality of data presentation is low, and similarly how the seizures are measured, and how they are tested statistically, does not convince me that this approach is working. See below further comments:

I was surprised by the choice of model. Although acute systemic injection of 4-aminopyridine (4-AP) is used as a model of epilepsy in the literature, it is not very common. Other models that are more commonly used are injections of pentylenetetrazole or kainic acid. The issue with 4-AP is the increased presence of inhibitory GABAergic activity that may not necessarily mimic true human epilepsy. See:

4-Aminopyridine induces a long-lasting depolarizing GABA-ergic potential in human neocortical and hippocampal neurons maintained in vitro M Avoli, P Perreault, A Olivier, JG Villemure - Neuroscience letters, 1988 –

4-aminopyridine-induced epileptiform activity and a GABA-mediated long-lasting depolarization in the rat hippocampus P Perreault, M Avoli - Journal of Neuroscience, 1992 –

Nonetheless, this is really a proof of principle study, so it could be valid to use this model. But the data thus far is not very convincing:

Authors' response: We sincerely thank the reviewer for your constructive suggestion. In this work, we focused on developing high-performance and biocompatible lead-free piezoelectric materials through multilevel structural engineering, as suggested by the editorial team and reviewers #3. Based on the newly developed lead-free materials, we employed two sandwich porous 1-3 composite-based piezoelectric ultrasonic energy harvesters (SP-1-3 PUEHs) with different resonant frequencies (PUEH-1: 1 MHz, PUEH-2: 3 MHz) and connected them inversely to a pair of rectifiers and transistors, respectively, and then programmed the triggered US (frequency and phase) to achieve biphasic stimulation. Furthermore, we utilized the 4-AP seizure model to demonstrate the feasibility and effectiveness of this lead-free ultrasound device in DBS. Aminopyridines are a medication family that has the ability to improve synaptic transmission. Therein, 4-aminopyridine (4-AP) has been utilized as a model for generalized seizures in studying antiseizure treatments for more than 50 years (*Acta Neurobiologiae Experimentalis* 2023, 83, 63-70). In previously reported studies, this model helps researchers study different drugs and physical treatments, such as electrical stimulation, for reducing or eliminating seizures (*Brain Stimulation* 2020, 13, 1387e1395; *Brain* 2018, 141; 2083–2097; *Frontiers in Neuroscience* 2019, 13, 677). The advantage of 4-AP model is that seizure-like events occurred continuously with sufficient regularity to allow a range of stimulation parameters to be explored in vivo compared to other acute models (*Nature Biomedical Engineering* 2023, 7, 559-575; *Nature Medicine* 2001, 7, 1063-1067). In the revised manuscript, we gave additional explanations for the choice of this model. We appreciate the reviewer's careful reminder about the limitations of the 4-AP model, which is constructive for building a more effective epilepsy model for clinical research. Since the priority of our work is the development of biphasic stimulator for DBS based on the high-performance piezoelectric materials, epilepsy treatment is just one way to study the application of the device. Meanwhile, your comments and those of other reviewers suggested that the focus of this study is not on epilepsy but should have been more on innovations in materials and devices. Therefore, we look forward to following your suggestion to use a potentially better epilepsy model in future studies.

Revised text:

“To further investigate the DBS antiepileptic effects of the f-BUI system, a 4-aminopyridine (4-AP)-induced seizure model was established, which can generate table seizure-like events occurred conditions regarding evoked epileptiform burst activity morphology and frequency over several hours⁵⁷⁻⁵⁹.”

It is difficult for me to understand precisely what is being presented and how the stimulation affects overall brain activity. It would be important to show that a dataset where there is a clear effect of brain stimulation on brain activity. At the moment we only have the indirect effect on seizures. Basically, we would want to see the SHAM + US+f-BUI (non-seizure) group. Is there an effect on the electrographical data? This would also be helpful in dissociating whether the claimed effects on power reduction are specifically on seizures or on overall brain activity. Or just to prove that there is any effect at all of the stimulation. It would be useful to see plots of the electrographical data around the start of stimulation to see the effect on local field potential (LFP) activity. What does this high frequency stimulation do to overall brain activity is my query?

Authors' response: We are thankful for the reviewer's valuable suggestion. Herein, we provide the ECoG signals recorded while US+f-BUI stimulation in Sham group (**Supplementary Fig. 30a**). However, the amplitude of biphasic electrical stimulation artifacts is much larger than the local field potentials as shown in previous reports (*Clinical Neurophysiology* 2010, 121(8): 1227-32; *Experimental Neurology* 2013, 245 77-86), which made an interference on the LFP recording. Meanwhile, the ECoG signals recorded before and after US+f-BUI stimulation showed that the normal brain activity was similar with the signal before electrical stimulation (**Supplementary Fig. 30b**), demonstrating that US+f-BUI stimulation have no adverse effect on the normal brain activity, which was beneficial from the biosecurity afforded by biphasic electrical stimulation. In fact, the stimulation target is the anterior nucleus of the thalamus, an attractive target for the modulation of seizure networks, which can inhibit seizure by

Papez circuit under electrical stimulation. Since the ANT is involved in seizure propagation and spreading via corticothalamic connections, mammillary bodies, and the Papez circuit, stimulation of ANT is considered an effective approach to suppress abnormal neural activity originating from epilepsy (*Neurosurgical Review* 2019, 42 (2), 287-296; *Cerebral Cortex* 2022, 32 (24), 5530-5543; *Brain Stimulation* 2015, 8(6), 1049-1057). Therefore, we speculate that impact on the treatment of epilepsy was attributed to the high frequency electrical stimulation to ANT on the Papez circuit inhibiting abnormal neural activity by interfering with seizure synchronization. Given the focus of this manuscript on the design and development of lead-free materials and devices, we do not overemphasize the mechanism of our device's impact on the treatment of epilepsy.

Supplementary Fig. 30 | ECoG signals record in Sham group treated by US+f-BUI.

(a) Representative ECoG signals record during the US+f-BUI stimulation in Sham group. (b) ECoG signals record in the Sham+US+f-BUI group before and after US+f-BUI stimulation.

Revised text:

“In our experiments, electrocorticography (ECoG) signals recorded while US+f-BUI stimulation in Sham group (**Supplementary Fig. 30a**) showed that the amplitude of biphasic electrical stimulation artifacts is much larger than the local field potentials as shown in previous reports^{60,61}, which made an interference on the LFP recording. Meanwhile, the ECoG signals recorded before and after US+f-BUI stimulation showed

that the normal brain activity was similar with the signal before electrical stimulation (**Supplementary Fig. 30b**), demonstrating that US+f-BUI stimulation have no adverse effect on the normal brain activity, which was beneficial from the biosecurity afforded by biphasic electrical stimulation. Therefore, we recorded the ECoG signals before and after stimulation, and there were four groups: the sham group, the seizure group, the US alone (US) group, and the US-induced f-BUI (US+f-BUI) group.”

In the zoomed in time scale data across figure 4d of the electrographical data there is no scale bar. The end result showing decreased overall power in the spectrograms in d suggests that there is a decrease in overall power, but not complete return to normality or the conditions shown in sham. A better analysis could be to take shorter time periods. In acute models of epilepsy seizure-like events often come in bursts with breaks in the activity, we see this a bit in plots with long time periods of LFP data, with seizures seemingly waxing and waning. This on-and off type pattern is still seen in the US+f-BUI condition after stimulation, there is still an oscillatory rhythm which is not present in the sham conditions. Could the experimenters quantify instead the number of bursts of events? Or take something like a coastline measure? But, overall, it would be nice to see the immediate effect on seizures, not after stimulation, but as the stimulation comes on, in real time, with a clear demarcation of a before and after stimulation from the same continuous data. What is the transition between high seizure activity to low seizure activity after 15 minutes of stimulation? 15 minutes is a long time in terms of acute models, so try to break it down into shorter periods.

Authors’ response: We are grateful to the reviewer for the constructive suggestion. We are very sorry that we omitted some information, which caused the reviewer to have doubts about the result. Herein, the scale bar of zoomed electrographical data has been added in the **Fig. 4d**. In **Fig. 4g**, the integrated area under the PSD curve was used, representing the overall power or strength of the ECoG signals, to compare the difference among the groups (*Epilepsia* 2022; 63: 1630-1642; *Clinical Neurophysiology* 2015, 126(2): 237-48; *Journal of Neurophysiology* 2022, 128:1, 118-130). Furthermore, we performed 2-way ANOVA type statistical tests, and the results

confirmed that treatment with US+f-BUI was significantly lower than other groups and the strength of the ECoG signals was close to the Sham group. Although there was still a degree of oscillatory rhythm after US+f-BUI stimulation, the abnormal ECoG signals is significantly lower than the seizure group. The inhibitory effect of this device may be related to other stimulus parameters, and there are still many parameters, which should be to optimize to achieve more efficient effect of treatment, such as ultrasound intensity and frequency. But the results have confirmed that the current biphasic electrical stimulation induced by US+f-BUI can effectively reduce seizure severity.

LFPs are the weak potential change caused by neural activities in a local neural network, and their amplitude is usually below 50 μV , while the electrical stimulation artifacts are much larger than LFPs, which remains a challenge that how to deal with high frequency electrical stimulation artifacts in DBS studies (*Journal of Neural Engineering* 2021, 18(6): 066031; *Experimental Neurology* 2013, 245 77-86). Given the effect of stimulation artifacts on LFP recording, we recorded ECoG signals before and after stimulation to demonstrate the inhibitory effect of stimulation on epilepsy according to previous reports (*Cerebral Cortex* 2022, 00, 1-14; *Brain Stimulation* 2016, 9: 285-295). Furthermore, the seizure duration of 4-AP epilepsy model can offer the advantage of stable seizure-like events occurred conditions regarding evoked epileptiform burst activity morphology and frequency over several hours, which allows a range of stimulation parameters to be explored in vivo compared to other acute models (*Frontiers in Neuroscience* 2019, 13: 677; *Brain Research* 2008, 1187: 74-81). The 15-minute stimulation time was chosen to allow ample time for electrical stimulation to exert the maximum inhibitory effect on seizures. After electrical stimulation, the intensity of the epileptic signal changed from high to low, accompanied by behavioral changes, reflected in the reduction of Racine's score, as shown in the **Fig. 4h**.

According to the reviewer's recommendation, employing a shorter duration and recording while stimulating is beneficial for observing the direct effect of epilepsy suppression following stimulation. This provides us with new ideas for the next work,

and we will use the reviewer's advice to further examine the mechanism of action of our device in anti-epilepsy.

4e and f do not show a decrease in power. We do not see individual variability in power, there are no stats calculated to show that there are differences between the various groups. Could this be done in different spectral frequency bands? Similarly in 4g I do not understand the stats that are performed to reach the conclusion. You need to be able to show individuals and more than the comparison between the condition before and after the stimulation, you would want to compare across groups with a 2-way ANOVA type statistical tests. But overall I think it is critical to show a non-epilepsy stimulated group. The racine score experiment is okay but we need detailed stats and plots with individual animals shown. Furthermore, it would be great to see a progression of epileptic activity? Perhaps the authors could plot how quickly high racine scores are reached and compare this across groups.

Authors' response: Thank you for your constructive suggestion. We used the integrated area under the PSD curve, representing the overall power or strength of the ECoG signals, to compare the difference among the groups (*Epilepsia* 2022, 63: 1630-1642; *Clinical Neurophysiology* 2015, 126(2): 237-48; *Journal of Neurophysiology* 2022, 128: 118-130), which is enough to confirm the suppressive effect of US+f-BUI. In **Fig. 4g**, we performed 2-way ANOVA type statistical tests, and the results confirmed that the strength of the ECoG signals treated by US+f-BUI was significantly lower than other groups and was close to the Sham group, which confirmed biphasic electrical stimulation generated by US+f-BUI can effectively reduce seizure severity. Meanwhile, we made a modification of **Fig.4h** to show the Racine score of individual animals. We scored epileptic behavior in strict accordance with the rules of Racine score to reflect the progress of epileptic activity. Unfortunately, we did not accurately record how quickly we reached a high Racine score. However, after 4-AP infection, rats in all groups reached high Racine scores in almost the same time. Thanks again for the reviewer's comments. We will conduct a more detailed investigation on this indicator in the future.

Finally, 4J is not convincing at all. For me the images show different exposition levels. Can the authors quantify the number of cells? Can this be done in other structures in hippocampus and beyond? And, again, we would need a non-epileptic stimulated group to make sure there is no effect on seizures. Is there evidence in the literature that there is hippocampal sclerosis in this model?

Authors' response: We sincerely appreciate your constructive suggestions. The Nissl staining images of CA3 hippocampal regions were scanned with a digital panoramic scanner (WS-10, Wisleap, China) under the same parameters. 4-AP can induced intense and frequent epileptic discharges and damage in both the hippocampus and the cerebral cortex, resulting in the shrinking and loss of hippocampal cells in the CA1 and CA3, as well as hippocampal sclerosis (*Journal of Neurochemistry* 1999, 72(5), 2006-2014; *Brain Research* 2004, 1006(2):b225-232; *Journal of Neuroscience* 1997, 17(23): 9308-9314; *Neuroscience* 2000, 101(3): 547-61). The shrinking and loss of hippocampal cells are marker of hippocampal sclerosis. We focused on the shrinking of hippocampal cells and have quantified the number of cells in the CA3 region to demonstrate the effectiveness of the f-BUI system in anti-epilepsy, which indicated that treatment with US+f-BUI effectively reduced seizure severity. Meanwhile, we have added the CA3 region Nissl staining of a non-epileptic stimulated group, named as Sham+US+f-BUI, and the number of cells was quantitatively analyzed. It showed no significant difference from the Sham group, indicating that there was no negative impact of US+f-BUI on the normal state.

Revised text:

“Meanwhile, 4-AP induced intense and frequent epileptic discharges and damage in both the hippocampus and the cerebral cortex, resulting in the shrinking and loss of hippocampal cells in the CA1 and CA3, as well as hippocampal sclerosis⁶⁴⁻⁶⁷. As shown in **Fig. 4j** and **Supplementary Fig. 31**, the CA3 region Nissl staining and the number of cells were quantitatively analyzed. It showed no evident difference between the Sham group and Sham+US+f-BUI group, indicating that there was no negative impact

of US+f-BUI on the normal state. Furthermore, the architecture and cell number of the CA3 hippocampal regions in the US+f-BUI group was resembling to that in the sham group. Contrary to that, the shrinkage of hippocampal cells in the seizure and US groups increased due to the failure to suppress epilepsy in time.”

Fig. 4j Nissl staining in CA3 regions of hippocampus on day 30. Scale bar is 100 μ m.

Supplementary Fig. 31 | Quantification of the cells number in the CA3 region after different treatment ($n = 3$). Biologically independent samples; statistical significance was determined by ANOVA with Tukey’s multiple comparisons (** $p < 0.01$ and ns, not significant); data were presented as mean values \pm standard deviations (SD); error bars = SD.

One positive element is the survival. In a way this model is both acute and chronic as survival is measure in the model. Repeated measure ANOVAs should be used for the stats here and be well described.

Authors’ response: We are thankful for the reviewer’s valuable suggestion. We have performed data analysis of survival rate by the Kaplan-Meier estimate with log rank test. If seizures are not inhibited in time, the 4-AP-induced acute seizure model will

result in a significant fatality rate. Even if it is suppressed, the damage to brain tissue caused by epilepsy can also affect the survival of rats over time. As indicated by the reviewer, this model is both acute and chronic. As shown in Fig. 4i, the survival rate of the US+f-BUI group was higher than seizure group, demonstrating that the f-BUI system could effectively decrease the mortality of epileptic rats by inhibiting aberrant ECoG signals propagation.

Revised text:

“We further observed that the rate of death in the 4-AP rat epilepsy model was high if seizures were not inhibited in time. Of note, the survival rate of the US+f-BUI group was significantly higher than the seizure group, demonstrating that the f-BUI system could effectively decrease the mortality of epileptic rats by inhibiting aberrant ECoG signals propagation (Fig. 4i).”

Fig. 4i Survival rate of seizure rats induced by the 4-AP after different treatments on day 30 ($n = 12$). Biologically independent samples; statistical significance was determined by the Kaplan-Meier method with log-rank test; **** $p < 0.0001$, * $p < 0.05$; data were presented as mean values \pm standard deviations (SD); error bars = SD.

It is not clear to me the mechanism of how the anti-epileptic effect works, we need some incite into the circuitry involved. Why are the electrodes implanted in the locations stated, and what are they. I think recording is in the hippocampus while stimulation is in a more rostral structure, why these locations? How far reaching would

the stimulus be beyond the stimulated structure?

Authors' response: We are thankful for the reviewer's valuable suggestion. The stimulating electrode was implanted at the following coordinates: AP: -1.44 mm, ML: ± 1 mm, DV: 6 mm, which was the position of anterior nucleus of the thalamus (ANT). ANT stimulation has been approved in the European Union and the United States as an alternative therapy for pharmaco-resistant epilepsy, which has been demonstrated to be effective in decreasing seizure frequency and remodeling abnormal brain function (*Epilepsia* 2017, 58 (S1), 80-84; *Brain Stimulation* 2015, 8(6): 1049-1057). The mechanism of ANT stimulation is highly related to Papez circuit. The ANT has extensive frontal and temporal cortical projections, which is a key component in the limbic circuit of Papez. Since the ANT is involved in seizure propagation and spreading via corticothalamic connections, mammillary bodies, and the Papez circuit, stimulation of ANT is considered to be an effective approach to suppress abnormal neural activity caused by epilepsy (*Neurosurgical Review* 2019, 42(2): 287-296.). In fact, the recording electrode is near the hippocampus while stimulation is in a more rostral structure. The recording electrode is stainless steel screws, and the recording site (AP: -3.5 mm, ML: ± 1.2 mm) is near the hippocampus according to the typical protocol, which can obtain intense and frequent epileptic discharges ECoG signals (*STAR Protocols* 2021, 2(4): 100981; *Epilepsy Research* 2019, 156: 106131). In this work, we focused on the dual-frequency ultrasound-induced biphasic electrical stimulation for seizure suppression to investigate the feasibility and effectiveness of this device in DBS. Thanks for the reviewer's suggestion. In future, we will focus on the detail research of stimulus effect on the other stimulated structure to furtherly reveal mechanism of biphasic electrical stimulation for seizure suppression.

Finally, I concur with the other reviewers that the exact value of this approach is not clear as current DBS stimulators may be easier to control and already quite efficient.

Authors' response: We are grateful to the reviewer for the constructive suggestion. In our work, we focused on the design of two newly developed sandwich porous 1-3 (SP-1-3) lead-free piezoelectric components, which are biocompatible and remarkably

enhanced energy harvesting performance through multilevel structural engineering. Previous piezoelectric ultrasound stimulators were mostly based on lead-based piezoelectric ceramics and can only deliver monophasic stimuli. However, biocompatibility is essential and a biphasic stimulus is usually required for biomedical stimulators, especially for some high-frequency stimulations, since its charge balance waveform reduces charge buildup and undesirable electrochemical reactions on the electrode surface. To achieve these, we employed two lead-free sandwich porous 1-3 composite-based piezoelectric ultrasonic energy harvesters (SP-1-3 PUEHs) with different resonant frequencies (PUEH-1: 1 MHz, PUEH-2: 3 MHz) and connected them inversely to a pair of rectifiers and transistors, respectively, and then programmed the triggered US (frequency and phase) to achieve biphasic stimulation. On the one hand, the biphasic electrical stimulation is aimed to solve the problem of monophasic and wired electrical stimulation, which allows charge balancing of the electrode, improving stimulation responses and minimizing tissue damage and corrosion of the electrodes during chronic stimulation. On the other hand, as a proof-of-concept, we investigated the biphasic electrical stimulation for seizure suppression, which demonstrated the feasibility and effectiveness of this device in DBS.

Even if current DBS stimulators may be easier to control and already quite efficient, the power supply for DBS, however, remains a fundamental challenge to be addressed. Conventional external power supplies use transcutaneous or percutaneous wires, which are inconvenient, easily damaged, and susceptible to infection, especially in long-term applications. Integration of the battery with the implant, but due to the limited lifespan of the battery, patients need to undergo regular surgery to replace the dead battery with a new one, which undoubtedly adds to the patient's physical pain and financial burden.

Our device provides a possible solution based on wireless piezoelectric ultrasound technology and innovatively solves the disadvantages of single-phase electrical stimulation of traditional piezoelectric nanogenerator, realizing controllable biphasic electrical stimulation in a simple way. Meanwhile, compared with other wireless energy

transfer technologies (e.g., electromagnetic field), ultrasound converting mechanical energy into electrical energy is a early stage but promising technology that may offer better performance for small (mm-scale) and deep (several cm) implants in future because ultrasound can achieve better spatial resolution and longer travel depth in the tissue.

Based on the reviewer's comment, we have added a related discussion (for example, the control software and hardware optimization, the development of bioadhesive external transducers) at the end of the manuscript to further improve the current implant to make it more stable, efficient, and easier to operate.

REVIEWERS' COMMENTS

Reviewer #2 (Remarks to the Author):

Although authors did not fully address my comments, I appreciated their efforts on this work. I recommend this paper in this form to publish on Nature communications.

Reviewer #3 (Remarks to the Author):

The authors have significantly revised the introduction and satisfactorily addressed my concerns. I have a few minor suggestions (no need to return the manuscript to me):

1. Introduction "which will undoubtedly add to the physical pain and financial burden of patients".
Simply: "poses additional risk and a healthcare burden."
2. It is unclear what is meant by "arithmetic circuits". Suggestion "analog and digital circuits".
3. "Longer travel depth". Suggestion: "greater penetration through soft tissues."
4. It may be helpful to explicitly point out the potential miniaturization benefits of ultrasound compared to conventional wireless power transfer.

Reviewer #4 (Remarks to the Author):

I am satisfied with the rebuttal and the new quantifications provided of this being a robust proof of concept of the method.

My only comment is that the following rebuttal section should be included in the text. Essentially saying there is some seizure reduction but likely not total as of yet:

Although there was still a degree of oscillatory rhythm after US+f-BUI stimulation, the abnormal ECoG signals is significantly lower than the seizure group. The inhibitory effect of this device may be related to other stimulus parameters, and there are still many parameters, which should be to optimize to achieve more efficient effect of treatment, such as ultrasound intensity and frequency. But the results have confirmed that the current biphasic electrical stimulation induced by US+f-BUI can effectively reduce seizure severity.

Otherwise I would suggest that this is accepted for publication.

Point-by-point response to the reviewers' comments on MS ID NCOMMS-23-49482B

Dear reviewers,

We deeply appreciate the effort that you have taken in re-reviewing our manuscript entitled "Lead-free dual-frequency ultrasound implants for wireless, biphasic deep brain stimulation" (NCOMMS-23-49482B). These comments and suggestions were of great help and encouragement to us, and we have revised the main text according to the suggestions.

These modifications have been made in the revised manuscript. Here, we have also included a point-by-point response to the reviewers's comments (the original comments are in **black** color and our responses are in **light blue**), as shown below:

Reviewer #1 (Remarks to the Author):

Although authors did not fully address my comments, I appreciated their efforts on this work. I recommend this paper in this form to publish on Nature communications.

Authors' response: We greatly appreciate once again the reviewer for all your constructive suggestions and for your recommendation to publish our paper.

Reviewer #2 (Remarks to the Author):

The authors have significantly revised the introduction and satisfactorily addressed my concerns. I have a few minor suggestions (no need to return the manuscript to me):

Authors' response: Again, we highly appreciate the constructive suggestions of this reviewer on our manuscript. These suggestions improved our manuscript a lot. All revisions are listed below:

1. Introduction "which will undoubtedly add to the physical pain and financial burden of patients". Simply: "poses additional risk and a healthcare burden."

Authors' response: We are thankful for the reviewer's suggestion. We have replaced this description with "poses additional risk and a healthcare burden" in the Introduction (lines 36-37, page 3).

2. It is unclear what is meant by "arithmetic circuits". Suggestion "analog and digital circuits".

Authors' response: Thanks for the suggestion. We have replaced "arithmetic circuits" with "analog and digital circuits" (lines 71-72, page 4).

3. "Longer travel depth". Suggestion: "greater penetration through soft tissues."

Authors' response: We are thankful for the reviewer's suggestion. We have replaced "longer travel depth" with "greater penetration through soft tissues" in the revised manuscript (lines 47, page 3).

4. It may be helpful to explicitly point out the potential miniaturization benefits of ultrasound compared to conventional wireless power transfer.

Authors' response: We thank the reviewers for the suggestion. In the revised manuscript, we have added the potential benefits of ultrasound in terms of miniaturization compared to conventional wireless power transfer (lines 47-48, page 3).

Reviewer #3 (Remarks to the Author):

I am satisfied with the rebuttal and the new quantifications provided of this being a robust proof of concept of the method.

My only comment is that the following rebuttal section should be included in the text. Essentially saying there is some seizure reduction but likely not total as of yet:

Although there was still a degree of oscillatory rhythm after US+f-BUI stimulation, the abnormal ECoG signals is significantly lower than the seizure group. The inhibitory effect of this device may be related to other stimulus parameters, and there are still many parameters, which should be to optimize to achieve more efficient effect of treatment, such as ultrasound intensity and frequency. But the results have confirmed that the current biphasic electrical stimulation induced by US+f-BUI can effectively reduce seizure severity.

Otherwise I would suggest that this is accepted for publication.

Authors' response: Once again, we sincerely thank the reviewer for all the constructive suggestions and for the recommendation to publish our paper.

We agree with the reviewer's comment. Following the suggestion, we have added the above discussion to the main text in the revised manuscript (lines 333-340, page 13). The revised text is copied below:

“Consequently, although there was still a degree of oscillatory rhythm after US+f-BUI stimulation, the abnormal ECoG signals is significantly lower than the seizure group. The inhibitory effect of this device may be related to other stimulus parameters, and there are still many parameters, which should be to optimize to achieve more efficient effect of treatment, such as US intensity and frequency. However, the results have confirmed that the current biphasic electrical stimulation induced by US+f-BUI can effectively reduce seizure severity, showing promising DBS applications in the

treatment of epilepsy and other neurological disorders through further research in the future.” (lines 333-340, page 13)